# REINFORCED PREFERENCE OPTIMIZATION FOR RECOMMENDATION

## ABSTRACT

Recent breakthroughs in large language models (LLMs) have fundamentally shifted recommender systems from discriminative to generative paradigms, where user behavior modeling is achieved by generating target items conditioned on historical interactions. Yet current generative recommenders still suffer from two core limitations: the lack of high-quality negative modeling and the reliance on implicit rewards. Reinforcement learning with verifiable rewards (RLVR) offers a natural solution by enabling on-policy sampling of harder negatives and grounding optimization in explicit reward signals. However, applying RLVR to generative recommenders remains non-trivial. Its unique generation space often leads to invalid or repetitive items that undermine sampling efficiency, and ranking supervision is sparse since most items receive identical zero rewards. To address these challenges, we propose **Reinforced Preference Optimization for Recommendation** (**ReRe**), a reinforcement-based paradigm tailored to LLM-based recommenders, an important direction in generative recommendation. ReRe incorporates constrained beam search to improve sampling efficiency and diversify hard negatives, while augmenting rule-based accuracy rewards with auxiliary ranking rewards for finer-grained supervision. Extensive experiments on three real-world datasets demonstrate that ReRe consistently outperforms both traditional and LLM-based recommenders in ranking performance. Further analysis shows that ReRe not only enhances performance across both base and SFT-initialized models but also generalizes robustly across different backbone families and scales. Beyond empirical gains, we systematically investigate the design space of RLVR in recommendation across generation, sampling strategy, reward modeling, and optimization algorithm, offering insights for future research. Our codes are available at https://anonymous.4open.science/r/ReRe-E1B0.

## 1 INTRODUCTION

Recommender systems aim to learn users' ranking preferences over items, typically by contrasting target items against negative ones derived from historical user interactions (Rendle et al., 2012; Wu et al., 2021). With recent breakthroughs of large language models (LLMs) (OpenAI, 2023; Dubey et al., 2024; DeepSeek-AI et al., 2025; 2024), the field is witnessing a fundamental paradigm shift from discriminative recommenders to generative recommenders (Rajput et al., 2023; Zheng et al., 2023; Liao et al., 2024b; Zhai et al., 2024; Han et al., 2025). One promising direction in generative recommenders leverages the rich user behavior intelligence encoded in LLMs from large-scale web corpora, enabling LLMs to serve as recommenders (Bao et al., 2023b; Chen et al., 2024). In the paradigm of LLM-based recommender, a user's historical interactions within a predefined item corpus are reformatted into textual prompts, and the target item title or token is generated as the output. This process is typically realized either through supervised fine-tuning (SFT) (Bao et al., 2023a; Bismay et al., 2025), or through offline preference fine-tuning methods such as DPO (Rafailov et al., 2023), which rely on implicit rewards to inject ranking information by explicitly introducing negative items.

However, existing generative recommenders face two fundamental limitations: the modeling of high-quality negatives remains insufficient, and implicit rewards create a gap between generation likelihood margin and true user preferences, which weakens ranking supervision. Specifically, current generative recommenders either omit explicit negative modeling during training (Rajput et al., 2023;

Figure 1: Comparison of negative item source across different post-training methods. Notably, DPO samples random negative items while self-play DPO samples negative items from the frozen reference model $\pi_{\text{ref}}$. In contrast, RLVR directly samples negative items from the model $\pi_\theta$ being updated.

Zheng et al., 2023) or rely on negatives sampled from random distributions or frozen reference policies (Chen et al., 2024; Gao et al., 2024), as illustrated in Figure 1. Such a mismatch between the negative sampling distribution and the evolving generation distribution of the model often results in exposure to easy negatives, thereby limiting the model's discriminative capacity. Moreover, since the optimization objective is guided by implicit rewards derived from relative likelihoods rather than explicit user preferences, the training process is prone to reward hacking (Gao et al., 2023a; Rafailov et al., 2024), where reward scores improve while actual recommendation quality deteriorates.

Inspired by recent advancements in reinforcement learning with verifiable rewards (RLVR) (DeepSeek-AI et al., 2025; Liu et al., 2025b; Yue et al., 2025; Zheng et al., 2025), we find that RLVR provides a natural solution to the limitations of existing generative recommenders that rely on SFT followed by preference tuning. It enables on-policy sampling of higher-quality negatives while leveraging verifiable rewards to narrow the gap between implicit reward and true preference signals (Xie et al., 2025; Yu et al., 2025). However, applying RLVR to generative recommendation remains a non-trivial problem. Unlike open-ended language generation, recommendation introduces distinct challenges that complicate the adaptation of RLVR, mainly in two aspects:

- **Unique generation space.** In generative recommendation, the valid item space is much narrower than in open-ended language generation. If decoding is not explicitly constrained, the model frequently produces invalid items that are nonexistent in the item corpus (Bao et al., 2024). Furthermore, the constrained generation space leads to a steep token probability distribution, which makes the model prone to repeatedly generating the same items across multiple samples (*cf.* Section 4.2). As a result of both invalid and repetitive generations, the model encounters difficulties in both inference and sampling: standard sampling strategies frequently yield redundant items, substantially reducing sampling efficiency during training. Consequently, at each training step, the model is exposed to only a very limited range of negative samples, which restricts the diversity of ranking signals and results in inefficient optimization.

- **Sparse ranking supervision.** Rule-based reward modeling in RLVR functions as a binary correctness signal, assigning a reward of one to the correct item and zero to all others (Hu et al., 2025). In recommendation, however, user interactions are inherently sparse, and the vast majority of items in the corpus remain unobserved (Wei et al., 2024). As a result, nearly all sampled items are uniformly treated as negatives, each assigned an identical reward of zero. This leads to sampling groups where only the single target item is distinguished, while all other items collapse into the same reward value. Such sparsity provides only weak supervision signals during optimization and leaves the model without fine-grained ranking feedback.

Beyond these challenges, the adaptation of RLVR to recommendation remains largely underexplored, and the design space across its components has not been systematically investigated.

To this end, we conduct a comprehensive study of RLVR for generative recommendation, exploring its generation paradigm, sampling strategies, and reward modeling. Building on our insights, we propose Reinforced Preference Optimization for Recommendation (ReRe), a simple yet effective method that tailors RLVR to LLM-based recommenders, depicted in Figure 2. For generation, ReRe adopts constrained decoding by masking invalid tokens at each step, ensuring that only valid items are produced and fundamentally distinguishing the output space from that of open-ended language tasks. For sampling, ReRe employs beam search (Zheng et al., 2024) to efficiently generate diverse candidate items in a single pass, guaranteeing both sampling efficiency and exposure to informative negatives. For reward modeling, ReRe augments rule-based accuracy rewards with ranking rewards, which assign additional penalties to hard negatives according to their generation probabilities. Altogether,

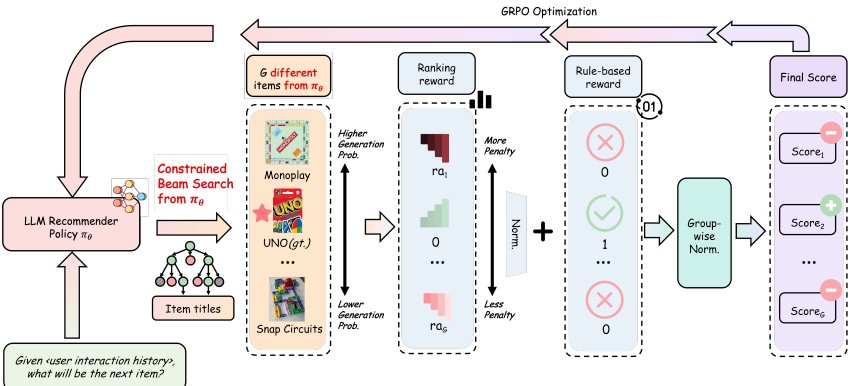

Figure 2: Framework of ReRe, where *gt.* means ground truth and $ra_k = -\frac{1}{\log(1+k)}$ . At each iteration, ReRe first **samples $G$ mutually distinct items directly from the training policy $\pi_\theta$ given the prompt containing user interactions via constrained beam search**. The generated items are then assigned **the rule-based reward plus the ranking reward**, from which the advantages are computed to update the model via GRPO.

ReRe offers a principled yet practical way to bring RLVR into generative recommendation, enhancing both the quality of negatives and the fidelity of ranking signals.

Extensive experiments on three real-world datasets demonstrate that ReRe consistently outperforms both traditional and LLM-based recommenders, yielding substantial improvements in ranking performance. Given that reinforcement-based paradigms for recommendation remain largely unexplored, we further conduct a systematic study of design choices across generation, sampling, reward modeling, and optimization, offering insights and references for future research. Finally, additional analyses show that ReRe not only boosts performance across both base and SFT models but also generalizes robustly across different backbone families and scales.

## 2 PRELIMINARY

In this section, we first formulate the generative recommendation task. Then we introduce the existing post-training methods for LLM-based recommenders, and discuss the issues those methods entail.

### 2.1 TASK FORMULATION

Let $\mathcal{H}_u = \{i_1, i_2, \cdots, i_n\}$ denote the interaction history of user $u$, where items are ordered chronologically. Given the history $\mathcal{H}_u$, the LLM-based recommenders $\pi_\theta$, where $\theta$ represents the parameters, are required to directly generate the target item $i_t$ that satisfies the preferences of user $u$ from the item set $\{i_k\}_{k=1}^N$, wherein $N$ is the number of items. To constrain the model's output to valid item titles, we adopt the constrained token generation strategy during generation, following previous works (Bao et al., 2024). Please refer to Appendix A.2 for the details of constrained token generation.

### 2.2 POST-TRAINING OF LLM-BASED RECOMMENDERS

In terms of post-training, SFT and DPO are two prevalent methods to fine-tune pretrained LLMs for recommendation. In SFT, training data are built by pairing the prompt $x_u$ bearing the interaction history of user $u$ with the textual title $e_t$ of the target item $i_t$. Subsequently, LLM-based recommenders are fine-tuned with the language-modeling loss:

$$\max_{\pi_\theta} \sum_{(x_u, e_t)} \sum_{j=1}^{|e_t|} \log \pi_\theta(e_{t,j}|x_u, e_{t,<j}), \tag{1}$$

where $|e_t|$, $e_{t,j}$ and $e_{t,<j}$ represent the number of tokens in $e_t$, the $j$-th token of $e_t$ and tokens preceding $e_{t,j}$, respectively. $\pi_\theta(e_t|x_u)$ means the output probability of item title $e_t$ given prompt $x_u$.

However, there is a gap between language modeling and the ranking preference alignment essence of recommendation. Drawing inspiration from the success of direct preference alignment methods (Rafailov et al., 2023; Azar et al., 2024; Amini et al., 2024), some research concentrates on adapting DPO to recommendation (Chen et al., 2024; Gao et al., 2024). Instead of solely considering target items like equation 1, direct preference alignment simultaneously suppresses the influence of negatives by enlarging the implicit reward gap between positive and negative items with the following loss:

$$\mathcal{L}_{\text{DPO}} = -\mathbb{E}_{(x_u, e_t, e_d)} \left[ \log \sigma \left( \beta \log \frac{\pi_\theta(e_t|x_u)}{\pi_{\text{ref}}(e_t|x_u)} - \beta \log \frac{\pi_\theta(e_d|x_u)}{\pi_{\text{ref}}(e_d|x_u)} \right) \right]. \tag{2}$$

Here $\pi_{\text{ref}}$ and $e_d$ represent the reference model and the title of the sampled negative item, respectively. Although preference alignment methods explicitly model ranking information, negative items are often sampled at random (Chen et al., 2024), which provides LLMs with limited supervision signals. Self-play methods alleviate this issue by sampling the negative items from the reference model $\pi_{\text{ref}}$ for DPO training (Liao et al., 2024a; Gao et al., 2024). However, the discrepancy between the distribution of negative items and the distribution of the model being updated remains.

## 2.3 GRPO

Group Relative Preference Optimization (GRPO), introduced in Shao et al. (2024), has recently gained significant attention as a representative algorithm in reinforcement learning with verifiable rewards (RLVR). Despite its success in general reasoning tasks, the adaptation of RLVR to recommendation remains under-explored. A natural way to bridge this gap is to reinterpret its groupwise preference optimization within the recommendation formulation in Section 2.1, where candidate item titles are treated as generations and evaluated against the target item. Concretely, at each iteration, the LLM-based recommender $\pi_\theta$ generates a group of $G$ candidate item titles $\{e_k\}_{k=1}^G$ for the same user prompt $x_u$, and a rule-based reward is applied to assess each candidate with respect to the target item:

$$R_{\text{rule}}(e_k, e_t) = \begin{cases} 1, & e_k = e_t \\ 0, & \text{otherwise} \end{cases}, \tag{3}$$

To compute token-level advantages, GRPO avoids training a separate value model as in PPO (Schulman et al., 2017), and instead normalizes the rewards within each group:

$$\hat{A}_{k,j} = \frac{r_k - \text{mean}(\{r_k\}_{k=1}^G)}{\text{std}(\{r_k\}_{k=1}^G)}, \tag{4}$$

wherein $r_k$ represents $R_{\text{rule}}(e_k, e_t)$ and $\hat{A}_{k,j}$ denotes the advantage of the $j$-th token in generation $e_k$. Finally, the model is optimized with the clipped objective:

$$\mathcal{J}_{\text{GRPO}}(\theta) = \mathbb{E}_{x_u \sim D, \{e_k\}_{k=1}^G \sim \pi_\theta(e|x_u)} \left[ \frac{1}{G} \sum_{k=1}^G \frac{1}{|e_k|} \sum_{j=1}^{|e_k|} \left\{ \min \left[ \frac{\pi_\theta(e_{k,j}|x_u, e_{k,<j})}{\pi_{\text{ref}}(e_{k,j}|x_u, e_{k,<j})} \hat{A}_{k,j}, \right. \right. \right.$$
$$\left. \left. \left. \text{clip} \left( \frac{\pi_\theta(e_{k,j}|x_u, e_{k,<j})}{\pi_{\text{ref}}(e_{k,j}|x_u, e_{k,<j})}, 1-\varepsilon, 1+\varepsilon \right) \hat{A}_{k,j} \right] - \beta \mathbb{D}_{\text{KL}}[\pi_\theta||\pi_{\text{ref}}] \right\} \right]. \tag{5}$$

Recently, numerous extensions of GRPO have been proposed, such as DAPO and GSPO (Yu et al., 2025; Zheng et al., 2025), which further enhance stability and efficiency. These explorations are largely parallel to our work; for completeness, we also explore the effectiveness of some representative variants in recommendation, and report their results in Appendix E.

Naturally, reinforcement-based paradigm addresses DPO's suboptimal negative sampling limitations by sampling negatives directly from the evolving policy and grounding optimization in explicit verifiable rewards rather than implicit likelihood margins. However, its adaptation to recommendation remains non-trivial, introducing challenges such as inefficient sampling and sparse rewards while also opening a broad design space, which we detail next.

## 3 METHODOLOGY

Adapting RLVR to recommendation is not straightforward, as it raises two key challenges in sampling and reward design. First, unlike open-ended language generation, the output space of recommendation

is constrained to valid item titles. It is much narrower and makes repeated sampling from the same prompt prone to severe duplication, which in turn leads to low sampling efficiency (*cf.* Section 4.2). To mitigate this, we incorporate dynamic sampling (Yu et al., 2025) and constrained beam search to improve sampling efficiency and expose the model to more informative negative items.

On the other hand, while ranking ability is essential for recommendation, the rule-based reward assigns zero to all negatives, providing only coarse-grained feedback. This exposes a broader challenge: reward design in recommendation remains under-explored, yet holds significant potential for richer supervision. To move beyond binary correctness, we propose an auxiliary ranking reward that penalizes harder negatives with lower scores, and further investigate the potential of dense rewards such as semantic and collaborative signals.

### 3.1 SAMPLING STRATEGY

A key challenge in adapting RLVR to recommendation lies in the low sampling efficiency caused by the narrow item space: repeatedly sampling from the same prompt often yields duplicate items, reducing the diversity of sampled negatives. To quantify this, we define generation diversity as

$$\text{div}(\{e_k\}_{k=1}^G) = \frac{\text{uni}(\{e_k\}_{k=1}^G)}{G}. \tag{6}$$

where $\text{uni}(\{e_k\}_{k=1}^G)$ counts the number of unique items among $G$ generations. Higher diversity exposes the model to richer negative signals and thus improves ranking supervision.

To enhance sampling efficiency, we explore two complementary strategies. We first explore dynamic sampling, which over-generates candidates and then selects a subset that balances inclusion of the target item with maximizing negative diversity (details in Appendix A.1). However, this strategy requires generating substantially more samples, and its diversity also inevitably declines as training progresses. Motivated by these limitations, we ultimately adopt beam search as our sampling method. Beam search ensures that all sampled items are distinct, thereby improving efficiency and diversity. Following Bao et al. (2024), we remove length normalization to avoid amplification bias, with additional details reported in Appendix A.2.

### 3.2 REWARD DESIGN

Recommendation quality is usually measured by ranking metrics like NDCG, highlighting the importance of fine-grained ranking ability. However, the rule-based reward in GRPO only distinguishes the target item from all others, assigning 1 to the positive and 0 to every negative. This implicitly assumes that all negatives contribute equally during optimization and only injects coarse-grained ranking information into LLMs. To further corroborate our analysis, we provide a gradient study of GRPO in Appendix A.3.

Prior works have shown that emphasizing hard negatives improves the discriminative ability of recommendation models (Wu et al., 2021; Chen et al., 2024). Motivated by this insight, we introduce a ranking reward penalizing harder negatives more heavily. Specifically, for a generated negative item $e_k$, we compute its generation rank $\rho_k$ and assign a lower reward to negatives with higher-probability:

$$\hat{R}_{\text{rank}}(e_k, e_t) = \begin{cases} 0, & e_k = e_t \\ -\frac{1}{\log(\rho_k+1)}, & \text{otherwise} \end{cases}, \tag{7}$$

$$R_{\text{rank}}(e_k, e_t) = -\frac{\hat{R}_{\text{rank}}(e_k, e_t)}{\sum_{j=1}^G \hat{R}_{\text{rank}}(e_j, e_t)}, \tag{8}$$

Finally, the overall reward combines the standard rule-based reward and the ranking reward:

$$R(e_k, e_t) = R_{\text{rule}}(e_k, e_t) + R_{\text{rank}}(e_k, e_t). \tag{9}$$

This combined reward integrates binary correctness with ranking signals, providing richer supervision that enhances the model's ability to discriminate among negatives and improves ranking performance.

While the ranking reward provides finer-grained supervision than the binary rule-based reward, reward design in recommendation remains largely under-explored. Beyond binary correctness, recommendation scenarios naturally allow the use of richer dense signals that may further enhance alignment. For further exploration, we investigate two recommendation-specific dense rewards:

- **Semantic reward.** This reward measures the semantic similarity between the generated item and the target item, encouraging LLMs to capture semantic relatedness between items.
- **Collaborative reward.** To inject collaborative signals, each generated item is assigned the corresponding logit output by a traditional recommender system, reflecting collaborative filtering knowledge from historical user-item interactions.

These dense rewards offer a complementary perspective to binary correctness and allow us to examine whether richer supervision can substitute rule-based signals in RLVR for recommendation.

# 4 EXPERIMENTS

In this section, we aim to answer the following research questions:

- **RQ1:** How does ReRe perform compared with existing traditional recommendation models and LLM-based recommenders?
- **RQ2:** How do different design choices, such as sampling strategies, reward formulations, and optimization algorithms, affect the performance of ReRe?
- **RQ3:** How well does ReRe generalize across backbone models of different families and scales?
- **RQ4:** How sensitive is ReRe to critical hyperparameters, and what settings strike the best balance between efficiency and effectiveness?

**Datasets and Metrics.** We evaluate ReRe on three real-world datasets: *Toys and Games* and *Industrial and Scientific* from the Amazon Review Dataset[1], and *Yelp* Dataset[2]. For evaluation, we adopt Hit Ratio (HR@K) and Normalized Discounted Cumulative Gain (NDCG@K) as metrics. Further details are provided in Appendix B.1.

**Baselines.** Our baselines comprise two categories: (1) Traditional recommendation models, including GRU4Rec (Hidasi et al., 2016), Caser (Tang & Wang, 2018), SASRec (Kang & McAuley, 2018); (2) LLM-based recommendation models, including TIGER (Rajput et al., 2023), BigRec (Bao et al., 2023a), D$^3$ (Bao et al., 2024), S-DPO (Chen et al., 2024), SPRec (Gao et al., 2024). Please refer to Appendix B.2 for more details.

**Implementations.** For ReRe, we use a training batch size of 512, with learning rate set to $1e-5$, $\beta$ set to $1e-3$, and the number of generations $G$ set to 16. We train ReRe models from the vanilla Qwen2-0.5B model (Yang et al., 2024a) or the SFT model for 2 epochs on 8 NVIDIA H20 GPUs. Please check Appendix B.3 for more implementation details.[3]

## 4.1 OVERALL PERFORMANCE (RQ1)

Based on the results in Table 1, we derive the following observations:

- **ReRe achieves superior performance across various datasets and initial models.** ReRe consistently surpasses both traditional and LLM-based recommenders, with relative gains of 27.13%, 12.40%, and 8.95% on Amazon Toys, Amazon Industrial, and Yelp, respectively. It benefits from more informative negatives and richer, verifiable reward signals, enabling strong generality across both base and SFT-initialized models. Notably, starting reinforcement learning directly from a base LLM yields weaker results on Yelp, likely due to the domain-specific nature of local business reviews, which are underrepresented in pretraining. Introducing an SFT stage before reinforcement learning remedies this gap and leads to stronger performance.
- **Better negative sampling leads to stronger user preference alignment.** SFT methods (e.g., BigRec) lack explicit negatives and thus miss ranking signals, leading to suboptimal performance. D$^3$ and S-DPO prove that injecting ranking information improves performance, but off-policy

---

[1] https://cseweb.ucsd.edu/~jmcauley/datasets.html#amazon_reviews
[2] https://business.yelp.com/data/resources/open-dataset/
[3] Empirically, ReRe from the SFT model tends to converge after 1 epoch.

Table 1: Recommendation performance on three real-world datasets. The best performance is highlighted in boldface, while the second-best performance is underlined.

| Dataset | Models | HR@3 | NDCG@3 | HR@5 | NDCG@5 | HR@10 | NDCG@10 |
|---|---|---|---|---|---|---|---|
| Toys | GRU4Rec | 0.0148 | 0.0120 | 0.0204 | 0.0143 | 0.0312 | 0.0178 |
| | Caser | 0.0216 | 0.0177 | 0.0280 | 0.0203 | 0.0397 | 0.0241 |
| | SASRec | 0.0357 | 0.0295 | 0.0431 | 0.0326 | 0.0581 | 0.0374 |
| | TIGER | 0.0383 | 0.0305 | 0.0507 | 0.0356 | 0.0715 | 0.0423 |
| | BigRec | 0.0420 | 0.0363 | 0.0530 | 0.0408 | 0.0715 | 0.0468 |
| | $D^3$ | 0.0564 | 0.0477 | 0.0710 | 0.0537 | 0.0940 | 0.0612 |
| | S-DPO | 0.0534 | 0.0449 | 0.0662 | 0.0502 | 0.0897 | 0.0578 |
| | SPRec | 0.0570 | 0.0479 | 0.0693 | 0.0529 | 0.0920 | 0.0602 |
| | **ReRe-Base** | **0.0748** | **0.0626** | **0.0899** | **0.0688** | **0.1125** | **0.0762** |
| | **ReRe-SFT** | 0.0648 | 0.0557 | 0.0779 | 0.0610 | 0.0992 | 0.0679 |
| Industrial | GRU4Rec | 0.0638 | 0.0542 | 0.0774 | 0.0598 | 0.0999 | 0.0669 |
| | Caser | 0.0618 | 0.0514 | 0.0717 | 0.0555 | 0.0942 | 0.0628 |
| | SASRec | 0.0790 | 0.0700 | 0.0909 | 0.0748 | 0.1088 | 0.0806 |
| | TIGER | 0.0852 | 0.0742 | 0.1010 | 0.0807 | 0.1321 | 0.0908 |
| | BigRec | 0.0931 | 0.0841 | 0.1092 | 0.0907 | 0.1370 | 0.0997 |
| | $D^3$ | 0.1024 | 0.0911 | 0.1213 | 0.0989 | 0.1500 | 0.1082 |
| | S-DPO | 0.1032 | 0.0906 | 0.1238 | 0.0991 | 0.1524 | 0.1082 |
| | SPRec | 0.1081 | 0.0984 | 0.1231 | 0.1046 | 0.1525 | 0.1139 |
| | **ReRe-Base** | **0.1222** | **0.1079** | **0.1447** | **0.1171** | **0.1707** | **0.1256** |
| | **ReRe-SFT** | 0.1103 | 0.0974 | 0.1275 | 0.1045 | 0.1546 | 0.1113 |
| Yelp | GRU4Rec | 0.0151 | 0.0112 | 0.0255 | 0.0155 | 0.0433 | 0.0211 |
| | Caser | 0.0125 | 0.0089 | 0.0189 | 0.0116 | 0.0330 | 0.0161 |
| | SASRec | 0.0127 | 0.0095 | 0.0182 | 0.0117 | 0.0341 | 0.0167 |
| | TIGER | 0.0154 | 0.0113 | 0.0251 | 0.0152 | 0.0427 | 0.0209 |
| | BigRec | 0.0173 | 0.0131 | 0.0255 | 0.0164 | 0.0398 | 0.0210 |
| | $D^3$ | 0.0196 | 0.0146 | 0.0289 | 0.0184 | 0.0492 | 0.0249 |
| | S-DPO | 0.0172 | 0.0130 | 0.0264 | 0.0168 | 0.0467 | 0.0233 |
| | SPRec | 0.0192 | 0.0143 | 0.0293 | 0.0184 | 0.0489 | 0.0247 |
| | **ReRe-Base** | 0.0173 | 0.0126 | 0.0252 | 0.0159 | 0.0408 | 0.0210 |
| | **ReRe-SFT** | **0.0218** | **0.0166** | **0.0315** | **0.0206** | **0.0499** | **0.0265** |

negatives of common DPO introduce a gap between policy and sampling distributions. Although SPRec narrows this gap through self-play, the negative distribution remains misaligned with the evolving model. ReRe addresses this by adopting a fully on-policy strategy, generating hard and diverse negatives directly from the training policy and further distinguishing them through a ranking reward. This combination exposes the model to more informative negatives and provides fine-grained supervision, ultimately enhancing its discriminative ability.

• **The adaptation of LLMs to recommendation substantially enhances their performance.** LLM-based recommenders consistently outperform traditional models, benefiting from their powerful task-executing ability and extensive world knowledge(Wu et al., 2023). While the potential is evident, effectively adapting LLMs to recommendation — a ranking-oriented task fundamentally different from general language generation — remains a non-trivial challenge.

## 4.2 STUDY ON RERE (RQ2)

**Reward design.** We compare four rewards: the rule-based reward that assigns binary correctness, our proposed ranking reward that further penalizes harder negatives, a semantic reward based on *ada-v3-large* similarity, and a collaborative reward derived from *SASRec* prediction scores (*cf.* Section 3.2). As shown in Table 2, the ranking reward achieves the best performance. In contrast, dense proxy rewards such as semantic and

Table 2: Study of different reward designs on Industrial.

| Reward | H@5 | N@5 | H@10 | N@10 |
|---|---|---|---|---|
| Ranking | **0.1447** | **0.1171** | **0.1707** | **0.1256** |
| Rule | 0.1443 | 0.1134 | 0.1705 | 0.1220 |
| Semantic | 0.0569 | 0.0389 | 0.0587 | 0.0394 |
| Collaborative | 0.0540 | 0.0296 | 0.0889 | 0.0414 |

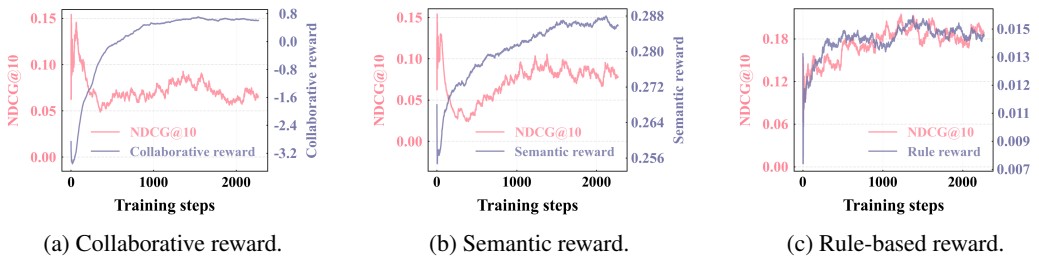

(a) Collaborative reward.  (b) Semantic reward.  (c) Rule-based reward.

Figure 3: Comparison among the consistency of three rewards and NDCG@10 metrics on Industrial.

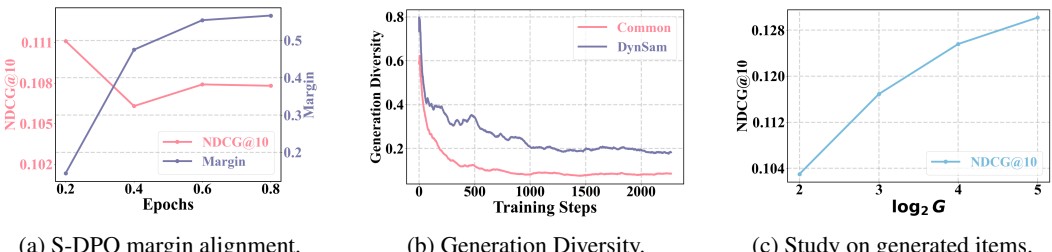

(a) S-DPO margin alignment.  (b) Generation Diversity.  (c) Study on generated items.

Figure 4: Study on Industrial: (4a) Reward margin and N@10 during S-DPO training. (4b) Diversity evolution of different sampling strategies. (4c) Effect of the number of generated items $G$ in ReRe.

collaborative are prone to reward hacking: as illustrated in Figure 3a and Figure 3b, their values continue to rise even when recommendation accuracy declines, revealing misalignment with the true objective. The rule-based reward is robust because of its verifiable construction, and enriching it with ranking information further improves NDCG@K by 3.11% on Amazon Industrial and 3.95% on Amazon Toys (*cf.* Table 8), confirming its effectiveness in enhancing discriminative ability.

To further contrast verifiable rewards with implicit rewards applied in DPO (Ave$_{\text{margin}}$ = $\frac{1}{|\mathcal{N}|} \sum_{i_d \in \mathcal{N}} \left[ \beta \log \frac{\pi_\theta(e_t|x_u)}{\pi_{\text{ref}}(e_t|x_u)} - \beta \log \frac{\pi_\theta(e_{i_d}|x_u)}{\pi_{\text{ref}}(e_{i_d}|x_u)} \right]$, $\mathcal{N}$ denotes the index set of sampled negatives), we analyze reward margins in S-DPO and observe similar misalignment trends (Figure 4a), highlighting the need for verifiable and better-aligned rewards in recommendation. Further settings and results are presented in Appendix D.2.

**Sampling strategy.** We compare three sampling strategies in ReRe: beam search, dynamic sampling, and common sampling. As shown in Table 3, beam search achieves the best results, followed by dynamic sampling, while common sampling performs worst. Interestingly, beam search requires fewer sampled generations yet still surpasses dynamic sampling, indicating its higher sampling efficiency.

Table 3: Study of sampling strategies on Industrial.

| Sampling | H@5 | N@5 | H@10 | N@10 |
|---|---|---|---|---|
| Beam | **0.1443** | **0.1134** | **0.1705** | **0.1220** |
| Dynamic | 0.1282 | 0.1038 | 0.1540 | 0.1122 |
| Common | 0.1218 | 0.0970 | 0.1447 | 0.1044 |

To better understand these differences, we analyze the evolution of generation diversity during training (Figure 4b). Both dynamic and common sampling exhibit a sharp decline in diversity as training proceeds, which limits the model's exposure to informative negatives. Although dynamic sampling partially mitigates this issue by retaining more candidates, the diversity drop remains significant. In contrast, beam search inherently maintains higher diversity, ensuring more varied and informative negatives for training, and thus emerges as a more effective strategy for recommendation. Please refer to Appendix D.1 for more results.

### 4.3 GENERALITY OF RERE (RQ3)

To evaluate the generality of ReRe, we compare it with D$^3$, a strong baseline, on the Amazon Industrial dataset using three additional backbone models of varying scales and families: Qwen2.5-1.5B (Yang

Table 4: Comparison of $D^3$ and ReRe across different models on Industrial.

| Backbones | Methods | H@3 | N@3 | H@5 | N@5 | H@10 | N@10 |
|-----------|---------|-----|-----|-----|-----|------|------|
| **Qwen2.5-1.5B** | $D^3$ | 0.1094 | 0.0967 | 0.1253 | 0.1032 | 0.1524 | 0.1120 |
| | ReRe | **0.1253** | **0.1091** | **0.1438** | **0.1167** | **0.1727** | **0.1261** |
| **Gemma-2B** | $D^3$ | 0.0984 | 0.0869 | 0.1158 | 0.0941 | 0.1478 | 0.1043 |
| | ReRe | **0.1200** | **0.1056** | **0.1352** | **0.1119** | **0.1663** | **0.1220** |
| **Qwen2.5-7B** | $D^3$ | 0.1017 | 0.0899 | 0.1171 | 0.0962 | 0.1432 | 0.1046 |
| | ReRe | **0.1227** | **0.1095** | **0.1428** | **0.1176** | **0.1679** | **0.1432** |

et al., 2024b), Gemma-2B (Mesnard et al., 2024), and Qwen2.5-7B. As summarized in Table 4, ReRe consistently surpasses $D^3$, achieving relative improvements of 13.52%, 18.28%, and 20.70% on the three backbones, respectively. These results demonstrate that ReRe generalizes well across both different architectures and model sizes. For Qwen2.5-7B, we report results with 8 generations due to resource constraints, yet ReRe still delivers substantial gains. Further experiments with alternative training objectives are provided in Appendix E.

### 4.4 ANALYSIS ON HYPERPARAMETERS (RQ4)

The number of generated items $G$ is a critical factor in ReRe, as it controls how many candidate items are produced at each step, thereby affecting both the diversity of exposed negatives and the granularity of ranking signals. To study its effect, we vary $G \in \{4, 8, 16, 32\}$ and train ReRe on Amazon Industrial. As shown in Figure 4c, increasing $G$ consistently improves recommendation performance, underscoring the potential of ReRe. This trend can be attributed to the greater diversity of items exposed to the model, together with the richer ranking signals introduced by beam search and the ranking reward. A complementary analysis on the effect of $\beta$ is provided in Appendix H.

## 5 LIMITATION

Although ReRe proves effective, this direction is still under-explored and entails several limitations. First, its performance scales with the number of generations, but we only explore up to 32 per step due to resource constraints, leaving higher settings for future work. Second, ReRe depends on the prior knowledge of the backbone, showing suboptimal results in domains underrepresented in pretraining (e.g., Yelp); while in-domain SFT alleviates this, it introduces extra cost, and more efficient transfer remains to be explored. Third, reinforcement learning is expected to offer stronger generalization, yet we do not investigate cross-domain recommendation or cold-start scenarios, which we leave for future study. Finally, while we believe ReRe has the potential to benefit the broader family of generative recommenders, in this work we primarily focus on the representative branch of LLM-based recommenders, leaving other directions such as semantic ID–based generative recommenders (Rajput et al., 2023; Zheng et al., 2023) for future exploration.

## 6 CONCLUSION

In this work, we introduced ReRe, a reinforcement-based post-training paradigm for LLM-based recommenders. ReRe addresses two key limitations of existing approaches: suboptimal negative sampling and imprecise implicit rewards. To further incentivize recommendation performance, we explore sampling strategies and reward designs, incorporating beam search for harder and more diverse negatives and ranking rewards for finer-grained, verifiable supervision. Extensive experiments on three real-world datasets show that ReRe consistently surpasses both traditional and LLM-based baselines, yielding substantial improvements in ranking performance. Further analyses demonstrate its robustness across base and SFT-initialized models, as well as its generality across different backbone families and optimization algorithms. Our work highlights the potential of reinforcement-based paradigms in recommendation, which remain largely under-explored. We hope ReRe serves as a foundation for future research on adapting reinforcement learning to recommendation.[4]

---

[4]The related works and the use of LLMs are presented in Appendix C and I, respectively.

ETHICS STATEMENT

Our paper mainly concentrates on tailoring the RLVR paradigm for recommendation and proposes a novel reinforcement training paradigm, ReRe, which specifically involves search strategies and reward functions custom-designed. We have thoroughly examined the potential social impacts of our methods and we think there is no ethical concerns within our work.

REPRODUCIBILITY

For better reproducibility, we provide the methodological details in Appendix A, and the implementation and experimental settings in Appendix B. Furthermore, our implementation is publicly available at https://anonymous.4open.science/r/ReRe-E1B0, which also contains the datasets and environment requirements.

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

# Appendix

---

**Algorithm 1** Dynamic Sampling

---

**Require:** The original item list $\mathcal{I}_O$ with $|\mathcal{I}_O| == \lfloor \frac{3G}{2} \rfloor$.
1: $\mathcal{I}_S = []$
2: **for** each $i \in \mathcal{I}_O$ **do**
3:    **if** $i$ is the ground truth **then**
4:       $\mathcal{I}_S$.append($i$)
5:       $\mathcal{I}_O$.remove($i$)
6:    **end if**
7:    **if** $|\mathcal{I}_S| == G$ **then**
8:       **return**
9:    **end if**
10: **end for**
11: **for** each $i \in \mathcal{I}_O$ **do**
12:    **if** $i$ not in $\mathcal{I}_S$ **then**
13:       $\mathcal{I}_S$.append($i$)
14:       $\mathcal{I}_O$.remove($i$)
15:    **end if**
16:    **if** $|\mathcal{I}_S| == G$ **then**
17:       **return**
18:    **end if**
19: **end for**
20: **if** $|\mathcal{I}_S| < G$ **then**
21:    $\mathcal{I}_S$.extend(RandomSampler($\mathcal{I}_O, G - |\mathcal{I}_S|$))
22: **end if**

---

# A METHODOLOY DETAILS

## A.1 DYNAMIC SAMPLING

Specifically, our recommendation-oriented dynamic sampling first generates $\lfloor \frac{3G}{2} \rfloor$ items, denoted as $\mathcal{I}_O$, and subsequently selects $G$ items for loss optimization, denoted as $\mathcal{I}_S$. The selection follows two principles: (1) The target item in $\mathcal{I}_O$ are first appended to the selected items $\mathcal{I}_S$ to strengthen supervision signals; (2) The remaining items are selected to maximize diversity of negatives, exposing models to broader training signals. Given the original item set $\mathcal{I}_O$, our recommendation-tailored dynamic sampling strategy is described in Algorithm 1.

## A.2 CONSTRAINED BEAM SEARCH

We leverage constrained beam search in the training and inference processes to ensure the generated items are valid and different. The constrained beam search can be divided into two parts: the constrained token generation and the beam search.

### A.2.1 CONSTRAINED TOKEN GENERATION

Let $\mathcal{E} = e_i, i = 1, 2, \cdots n$ denote the item title set, where $e_i = [e_{i,1}, \cdots e_{i,T_i}]$ denotes the token sequences of item title $e_i$ and $e_{i,T}$ is [EOS], indicating the end of sequences . We first build a hash map $h$ based on the item title set, which can be formulated as follows:

$$h(\phi) = \{t | \exists e \in \mathcal{E}, e \text{ starts with sequence } [t]\}, \tag{10}$$
$$h([t_1, \cdots, t_L]) = \{t | \exists e \in \mathcal{E}, e \text{ starts with sequence } [t_1, \cdots, t_L, t]\}, \tag{11}$$

wherein $\phi$ denotes the empty sequence.

Given the prompt $x$, suppose the model has already generated a token sequence $s$ ($s = \phi$ at the beginning of the generation). Let $\boldsymbol{f} = (f_1, \cdots, f_V)$ denote the logits of tokens, where $f_t$ refers to the logits of token $t$ and $V$ represents the size of the vocabulary. We can obtain the valid tokens conditioned on the generated tokens $s$, denoted by $\mathcal{V}_{\text{valid}} = h(s)$. Logits of all invalid tokens will be

masked, allowing only valid items to be generated, as described by the following formulas.

$$\hat{f}_t = f_t \cdot \mathbb{I}(t \text{ is valid}) - \text{MaskVal} \cdot \mathbb{I}(t \text{ is invalid}), \tag{12}$$

$$p(\cdot|x, s) = \text{Softmax}(\hat{f}_1, \cdots, \hat{f}_V). \tag{13}$$

Here $\mathbb{I}(\cdot)$ is the indicator function and MaskVal is a large negative value (*e.g.,* $-10^9$) assigned to the logits of invalid tokens, so that their probabilities approach zero after softmax.

### A.2.2 BEAM SEARCH

Let $B$ denote the beam width, which is the number of responses to be generated, and $\mathcal{M}$ denote the LLM. Let $\mathcal{B}^T$ stand for the $B$ generated sequences after the $T$-th beam search iteration. Given a prompt $x$, beam search perform the following iterations to generate $B$ different responses, until $T$ reaches the pre-defined maximum generated length.

1. Initialize $\mathcal{B}^0 = \phi$.

2. During the first iteration, beam search first calculates the predicted distribution over the first generated token $P(\cdot|x) = \mathcal{M}(x)$. Then, beam search obtains tokens with top $B$ generated probabilities, formulated as follows:

$$\mathcal{V}'_1 = \arg \operatorname*{top}_t B \ P(t|x), \tag{14}$$

$$\mathcal{B}^1 = \{[t]|t \in \mathcal{V}'_1\}. \tag{15}$$

Meanwhile, we maintain a hash function $g$ such that $g([t]) = \log P(t|x), \forall [t] \in \mathcal{B}^1$.

3. During the $T$-th ($T > 1$) iteration, beam search similarly calculates the distribution over the next token for each sequence $s \in \mathcal{B}^{T-1}$, we update $g$:

$$g([s, t]) = \log P(s, t|x) = \log P(t|x, s) + \log P(s|x) = \log P(t|x, s) + g(s), \quad \forall t \in \mathcal{V}, \tag{16}$$

where $\mathcal{V}$ denotes the whole vocabulary. Subsequently, $\mathcal{B}^T$ can be computed in the following way:

$$\mathcal{S}^T = \{[s, t]|s \in \mathcal{B}^{T-1}, t \in \mathcal{V}\}, \tag{17}$$

$$\mathcal{B}^T = \arg \operatorname*{top}_{s' \in \mathcal{S}^T} B \ g(s') \tag{18}$$

The process of beam search ensures that each $\mathcal{B}^T$ contains $B$ distinct generations. Let $L_{\max}$ denote the maximum length of responses. As a result, the final responses generated by beam search $B^{L_{\max}}$ will also be distinct.

### A.3 GRADIENT ANALYSIS

The gradient of the GRPO objective takes the following form (Shao et al., 2024) (with the weakly weighted KL term omitted):

$$\nabla_\theta \mathcal{J}_{\text{GRPO}}(\theta) = \mathbb{E}_{x_u \sim D, \{e_k\}_{k=1}^G \sim \pi_\theta(e|x_u)}$$

$$\frac{1}{G} \sum_{k=1}^G \frac{1}{|e_k|} \sum_{j=1}^{|e_k|} \left[ \hat{A}_{k,j} + \beta \left( \frac{\pi_{\text{ref}}(e_{k,j}|x_u, e_{k,<j})}{\pi_\theta(e_{k,j}|x_u, e_{k,<j})} - 1 \right) \right] \nabla_\theta \log \pi_\theta(e_{k,j}|x_u, e_{k<j}). \tag{19}$$

As shown in equation 19, the gradient weight scales with $\hat{A}_{k,j}$, which in turn depends on the assigned reward. By assuming equal rewards for all negatives, this formulation ignores their varying difficulty and thus provides only coarse-grained supervision.

### A.4 PROMPT DESIGN

The prompt design of ReRe is based on the implementations in Bao et al. (2024). The prompt template is illustrutaed in Figure 5, where {instruction} represents the task instruction, while {category},

Below is an instruction that describes a task, paired with an input that provides further context. Write a response that appropriately completes the request.

### Instruction:
{instruction}

### User Input:
The user has browsed the following {category}s before: {history}.

### Response:
{output}

Figure 5: Prompt template of ReRe.

1. Given a list of {category}s the user recently enjoyed, please write a new {category} that the user may buy.
2. Considering the {category}s that have recently captured the user's interest, kindly create a compilation of other {category}s that the user might have interest in prior to this.
3. Based on the user's current preference, please draft a list of potential {category}s they may have experienced beforehand.
4. Reflecting on the {category} the user has taken pleasure in recently, we request that you formulate a list of {category}s that may have preceded the user's current enjoyment.
5. In light of the recent gaming enjoyment expressed by the user, please assemble a list of {category} that could potentially include past titles the user has engaged with.
6. Taking into account the {category}s that have lately provided enjoyment to the user, please put together an inventory of {category}s the user might have explored previously.
7. Given the user's newfound enjoyment of a particular {category}, would you kindly generate a roster of other {category}s that might resonate with their past gaming experiences?
8. In response to the user's recent fondness for a specific {category}, we seek your assistance in listing possible {category}s the user may have delighted in earlier.
9. With respect to the {category}s currently enjoyed by the user, please compile a suggestive list of {category}s they may have played in the past.
10. Bearing in mind the {category}s that the user has recently been enthralled by, please construct a catalog of other {category}s that the user potentially partook in beforehand.
11. In relation to the user's recent entertainment with a given {category}, it would be appreciated if you could curate a list of {category}s that might form part of the user's previous gaming history.

Figure 6: Instruction set used to construct prompts of ReRe.

{history} and {output} stand for the item category, user interaction history and the target item title, respectively. Following Llara (Liao et al., 2024b), the task instruction in each prompt is randomly sampled from the instruction set in Figure 6 during both training and inference processes for training stability. One prompt example is provided in Figure 7.

## B  EXPERIMENTAL SETTINGS

### B.1  DATASETS

We conduct extensive experiments on three real-world datasets, including two from Amazon Review data[5]: **Toys_and_Games** and *Industrial_and_Scientific*, as well as one from Yelp [6]. Considering the high resource consumption of LLM training, we truncate the datasets following (Bao et al., 2023a; 2024). Specifically, during data preprocessing, we first filter the items and users with too few interaction records (fewer than 5). Subsequently, for the Amazon Toys dataset, data samples with interaction time between October 2016 and November 2018 are selected. For the Amazon Industrial dataset, since it is smaller than the other two datasets, we retain the interaction data from October 1996 to November 2018. For Yelp, we select the interactions from the year of 2021. Besides, we set the maximum length of interaction sequences to 10 for all the models. The truncated dataset is partitioned chronologically into training, validation, and test sets with an 8:1:1 ratio. The statistics of the training datasets are presented in Table 5.

---

[5]https://cseweb.ucsd.edu/~jmcauley/datasets.html#amazon_reviews
[6]https://business.yelp.com/data/resources/open-dataset/

> Below is an instruction that describes a task, paired with an input that provides further context. Write a response that appropriately completes the request.
>
> ### Instruction:
> Given a list of toys and games the user recently enjoyed, please write a new toy or game that the user may buy.
>
> ### User Input:
> The user has browsed the following toys and games before: "LEGO Elves The Elves' Treetop Hideaway 41075", "Ty Beanie Boo Scooter ".
>
> ### Response:
> "LEGO CITY Garbage Truck 60118 ".

Figure 7: Prompt example used in ReRe.

Table 5: Statistics of datasets, where the numbers in "Train", "Valid" and "Test" denote the numbers of interactions in the training, validation and testing datasets, respectively.

| Dataset | Users | Items | Density | Interactions | Train | Valid | Test |
|---|---|---|---|---|---|---|---|
| Toys | 22,158 | 11,251 | 0.065% | 140,943 | 112,754 | 14,094 | 14,095 |
| Industrial | 7,694 | 3,686 | 0.187% | 45,324 | 36,259 | 4,532 | 4,533 |
| Yelp | 10,635 | 8,785 | 0.115% | 96,372 | 77,097 | 9,637 | 9,638 |

## B.2 BASELINES

We compare our ReRe paradigm with several representative baselines, covering two categories: traditional recommendation models and LLM-based recommenders. The details of selected baselines are described below:

- **GRU4Rec** (Hidasi et al., 2016) leverages the GRU acitechture to model user interaction behavior.
- **Caser** (Tang & Wang, 2018) captures user preferences with horizontal and vertical convolutional operations of CNN.
- **SASRec** (Kang & McAuley, 2018) utilizes the multi-head self-attention mechanism to encode the interaction sequences for next item prediction.
- **TIGER** (Rajput et al., 2023) proposes a novel semantic ID representation for items and trains a transformer-based model to predict the semantic ID of the next item.
- **BIGRec** (Bao et al., 2023a) fine-tunes LLM-based recommenders with language modeling loss and modifies the decoding strategy to ensure the generated items are valid items.
- **D³** (Bao et al., 2024) modifies the decoding strategy of BIGRec by eliminating the length normalization and introducing logits from traditional models.
- **S-DPO** (Chen et al., 2024) incorporates a preference alignment phase after SFT, which extends the DPO algorithm to multi-negative scenarios.
- **SPRec** (Gao et al., 2024) utilizes the self-play philosophy by leveraging the negative items generated from the reference model.

## B.3 IMPLEMENTATION DETAILS

Traditional recommendation models are trained with the binary entropy loss and the Adam optimizer. The learning rate is searched in $[1e-2, 1e-3, 1e-4]$, and the weight decay is searched in $[1e-2, 1e-3, 1e-4, 1e-5, 1e-6]$. The training batch size is set to 1024. For TIGER, we utilize T5 (Raffel et al., 2020) as its backbone. For LLM-based recommenders, we adopt Qwen2-0.5B (Yang et al., 2024a) as the base model to reduce the computational overhead, and optimizes models with the AdamW optimizer (Loshchilov & Hutter, 2019). During fine-tuning, the SFT and preference alignment data are processed in batches of 128 and the reinforcement learning data are processed in batches of 512. We apply the learning rate $3e-4$ to SFT models, $1e-5$ for S-DPO, SPRec and ReRe with a cosine learning rate scheduler used. SFT models are trained for 10 epochs and the early stopping patience is

Table 6: Training costs of different methods.

|            | ReRe | S-DPO | BigRec [2] | SASRec [3] |
|------------|------|-------|------------|------------|
| Industrial | 20   | 23    | 1.2        | 0.042      |
| Toys       | 40   | 48    | 4.0        | 0.22       |
| Yelp       | 30   | 38    | 2.3        | 0.059      |

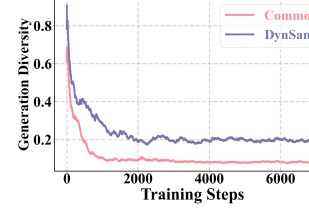

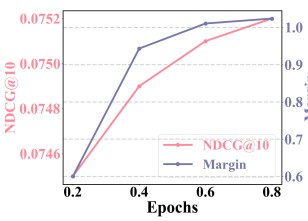

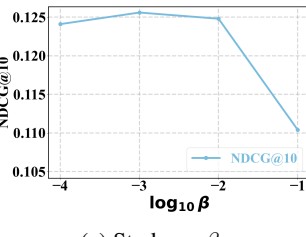

(a) Generation diversity.    (b) S-DPO margin alignment.    (c) Study on $\beta$.

Figure 8: (8a) Divsersity evolution of different sampling strategies on Toys. (8b) Reward margin and N@10 during S-DPO training on Toys. (8c) The influence of the value of $\beta$ on ReRe (Industrial).

set to 1. DPO models (S-DPO, SPRec) are trained for 1 epoch, while ReRe models are trained for 2 epochs. For $D^3$, $\alpha$ is searched in $[0.8, 0.9, 1.0]$. For DPO models, $\beta$ is set to $0.1$ and the number of negative items in S-DPO is set to 3. For ReRe, $\beta$ is set to $1e-3$ and the number of generated items is set to 16. When implementing GRPO algorithm, we estimate the KL divergence $\mathbb{D}_{\mathrm{KL}}$ by the following formula:

$$\mathbb{D}_{\mathrm{KL}}[\pi_\theta || \pi_{\mathrm{ref}}] = \frac{\pi_{\mathrm{ref}}(e_{k,j}|x_u, e_{k,<j})}{\pi_\theta(e_{k,j}|x_u, e_{k,<j})} - \log \frac{\pi_{\mathrm{ref}}(e_{k,j}|x_u, e_{k,<j})}{\pi_\theta(e_{k,j}|x_u, e_{k,<j})} - 1. \tag{20}$$

In addition, since the policy $\pi_\theta$ can deviate significantly from the initial policy during domain-specific knowledge injection, we dynamically update the reference policy in the training process following (Gorbatovski et al., 2024) to less restrict the model updating:

$$\pi_{\mathrm{ref}} \overset{\mathbf{sg}}{\longleftarrow} \alpha\pi_{\mathrm{ref}} + (1 - \alpha)\pi_{\mathrm{ref}_{\mathrm{prev}}}. \tag{21}$$

All experiments can be conducted on 8 NVIDIA H20 GPUs. The training costs of ReRe and baseline methods are in Table 6. [7] It can be concluded that although ReRe incurs additional overhead compared with SFT methods like BigRec, ReRe costs fewer GPU hours than direct preference alignment methods like S-DPO, which demonstrates ReRe's efficiency compared to other preference alignment methods.

### B.4 EVALUATION METRICS

We evaluate models with two broadly used metrics for recommendation: Hit Ratio (HR@K) and Normalized Discounted Cumulative Gain (NDCG@K). K is set to $3, 5$ and $10$ for comprehensive performance comparison. Besides, following (Bao et al., 2024), we apply a constrained token generation during the decoding of LLM-based recommenders. The length normalization elimination is adopted for $D^3$, S-DPO, SPRec and ReRe.

### B.5 DENSE REWARDS

For the semantic reward, we choose the text-embeddings-3-large (Neelakantan et al., 2022) released by OpenAI for text encoding to help LLMs grasp the semantic information. For collaborative rewards, each generated item is assigned the corresponding logit provided by a SASRec model (Kang & McAuley, 2018) as the collaborative reward.

---

[7]For fair comparison, the negative numbers of ReRe and S-DPO are both set to 15.

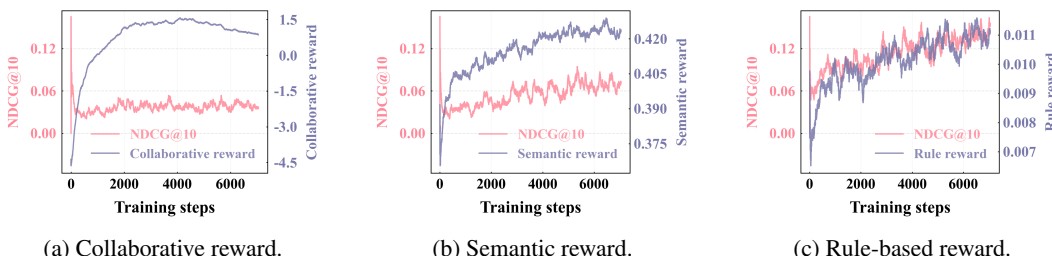

(a) Collaborative reward.     (b) Semantic reward.     (c) Rule-based reward.

Figure 9: Comparison among the consistency of three rewards and NDCG@10 metrics on Toys.

## C    RELATED WORK

### C.1    GENERATIVE RECOMMENDATION

The rapid advancement of LLMs paves the way for transforming the current recommendation from a discriminative paradigm to a generative paradigm Bao et al. (2023b); Zhou et al. (2025). In this context, a prominent line of research represents items using semantic IDs and leverages transformer-based models to generate candidate items, typically optimized with supervised fine-tuning (SFT) loss(Rajput et al., 2023; Hou et al., 2023; Zheng et al., 2023; Liu et al., 2025a). Both the encoding for semantic IDs (Wang et al., 2024) and the decoding strategy (Hou et al., 2025) play a crucial role in improving model performance in generating semantic IDs.

On the other hand, recent research has increasingly focused on employing LLMs directly as recommendation models. Preliminary research mainly focuses on directly using the in-context learning ability of LLMs (Brown et al., 2020) for zero-shot or few-shot recommendation tasks (Gao et al., 2023b; Liu et al., 2023). However, owing to the absence of domain-specific knowledge, untuned LLMs struggle to make more progress. To help LLMs understand recommendation tasks, recent works (Bao et al., 2023b;a; Zhang et al., 2023) introduce supervised fine-tuning for LLM training, which regards recommendation tasks as language modeling tasks. Moreover, researchers advance LLM-based recommendation tuning by supplementing collaborative information from a traditional model in different components, like token representations (Liao et al., 2024b; Zhang et al., 2025a; Li et al., 2023; Yang et al., 2023), LoRA modules (Kong et al., 2024), and the decoding phase (Bao et al., 2024). There are also works that endeavor to modify the training pipeline to bridge the gap between SFT and recommendation. These works append an extra user preference alignment stage to the SFT stage so that LLMs could better comprehend user preferences (Chen et al., 2024; Bai et al., 2024; Liao et al., 2024a; Gao et al., 2024). Moreover, there are also methods exploring RLVR for LLM-based recommender systems (Deng et al., 2025; Zhou et al., 2025; Zhang et al., 2025b; You et al., 2025). Compared with them, ReRe focuses on improving the model's ranking capability and incorporates the customized sampling strategy and reward mechanism for ranking information injection.

### C.2    REINFORCEMENT LEARNING FOR LLM

Reinforcement learning has been broadly used in LLM post-training (Zhao et al., 2023a). As a representative approach, reinforcement learning from human feedback (RLHF) (Ouyang et al., 2022; Bai et al., 2022; Ziegler et al., 2019) first fits a reward model from human preference datasets, and then trains an SFT model by reinforcement learning. Through RLHF, LLMs can be effectively aligned with human preferences. Besides, to spare high computational overhead and high instability of reinforcement learning, direct preference alignment methods are proposed (Zhao et al., 2023b; Rafailov et al., 2023; Ethayarajh et al., 2024). Among them, DPO (Rafailov et al., 2023) is more influential and numerous papers are proposed to make additional modifications to or conduct analysis on DPO (Azar et al., 2024; Meng et al., 2024; Yuan et al., 2024; Ren & Sutherland, 2025).

More recently, with the vast potential showcased by DeepSeek model series (DeepSeek-AI et al., 2025), increasing attention is paid to employing reinforcement learning along with the rule-based reward to stimulate the models' reasoning abilities. The relevant research includes providing more details about the reinforcement learning process for reasoning models (Yu et al., 2025; Xie et al.,

Table 7: Performance of different sampling strategies on Amazon Toys.

| Sampling | HR@5 | NDCG@5 | HR@10 | NDCG@10 |
|---|---|---|---|---|
| beam | **0.0866** | **0.0662** | **0.1086** | **0.0733** |
| dynamic | 0.0676 | 0.0502 | 0.0901 | 0.0574 |
| common | 0.0620 | 0.0434 | 0.0820 | 0.0498 |

Table 8: Performance of different reward designs on Amazon Toys, where "ranking" refers to the rule-based reward with the auxiliary ranking reward added.

| Reward | HR@5 | NDCG@5 | HR@10 | NDCG@10 |
|---|---|---|---|---|
| ranking | **0.0899** | **0.0688** | **0.1125** | **0.0762** |
| rule | 0.0866 | 0.0662 | 0.1086 | 0.0733 |
| semantic | 0.0370 | 0.0247 | 0.0556 | 0.0307 |
| collaborative | 0.0228 | 0.0129 | 0.0437 | 0.0197 |

2025; Hu et al., 2025) and proposing novel reinforcement learning algorithms (Hu, 2025; Zheng et al., 2025; Zhao et al., 2025).

## D  MODEL STUDY

This section presents the analysis results of ReRe on Amazon Toys. The analysis concentrates on two parts: the sampling strategy and the reward design.

### D.1  SAMPLING STRATEGY

The empirical results of different sampling strategies are in Table 7. Combined with the similar diversity drop phenomenon illustrated in Figure 8a, we can conclude that with more stable generation diversity, the beam search successfully further injects richer ranking information into LLM-based recommenders.

### D.2  REWARD DESIGN

Further performance comparison is in Table 7. Similar to the results on Amazon Industrial, the rule-based reward significantly surpasses the semantic reward and the collaborative reward on Amazon Toys. This improvement comparison substantiates the potential of the ranking reward defined in equation 8 on enhancing the model's fine-grained ranking ability by complementing the in-group sparsity with ranking-aware reward values.

The reward hacking analysis on the Amazon Toys is illustrated in Figure 9, where the similar misalignment between NDCG@10 performance metric and the reward values can be observed for both the semantic reward and the collaborative reward. Meanwhile, the rule-based reward aligns well with the recommendation performance. Regarding the implicit reward in S-DPO, Figure 8b shows that it aligns better for Amazon Toys than for Amazon Industrial. However, the observed changes in NDCG metrics are too minor to fully confirm the alignment of the implicit reward.

## E  STUDY OF OPTIMIZATION ALGORITHM

Recent research explores different modifications to GRPO objectives. In this part we replace the GRPO training objective with the following two representative algorithms:

- **DAPO** (Yu et al., 2025) decouples the higher and lower clipping range and introduces a token-level policy gradient loss, formulated as follows.

$$\mathcal{J}_{\text{DAPO}}(\theta) = \mathbb{E}_{x_u \sim D, \{e_k\}_{k=1}^{G} \sim \pi_\theta(e|x_u)} \left[ \frac{1}{G \sum_{k=1}^{G} |e_k|} \sum_{k=1}^{G} \sum_{j=1}^{|e_k|} \left\{ \min \left[ \frac{\pi_\theta(e_{k,j}|x_u, e_{k,<j})}{\pi_{\text{ref}}(e_{k,j}|x_u, e_{k,<j})} \hat{A}_{k,j}, \right. \right. \right.$$

(22)

$$\left. \left. \left. \text{clip}\left( \frac{\pi_\theta(e_{k,j}|x_u, e_{k,<j})}{\pi_{\text{ref}}(e_{k,j}|x_u, e_{k,<j})}, 1 - \varepsilon_{\text{low}}, 1 + \varepsilon_{\text{high}} \right) \hat{A}_{k,j} \right] \right\} \right].$$

(23)

- **GSPO** (Zheng et al., 2025) defines the importance ratio based on sequence likelihoods, ensuring the alignment between sequence-leven rewarding and optimization:

$$\mathcal{J}_{\text{GSPO}}(\theta) = \mathbb{E}_{x_u \sim D, \{e_k\}_{k=1}^{G}} \left[ \frac{1}{G} \sum_{k=1}^{G} \min(s_k(\theta)\hat{A}_k, \text{clip}(s_k(\theta), 1 - \varepsilon, 1 + \varepsilon)\hat{A}_k \right], \quad (24)$$

$$s_k(\theta) = \left( \frac{\pi_\theta(e_k|x_u)}{\pi_{\text{ref}}(e_k|x_u)} \right)^{\frac{1}{|e_k|}}. \quad (25)$$

The evaluation results on the Amazon Industrial dataset are in the following table:

|  | H@3 | N@3 | H@5 | N@5 | H@10 | N@10 |
|---|---|---|---|---|---|---|
| ReRe (GRPO) | 0.1253 | 0.1091 | 0.1438 | 0.1167 | **0.1727** | 0.1261 |
| ReRe (DAPO) | 0.1238 | 0.1083 | 0.1418 | 0.1158 | 0.1703 | 0.1256 |
| ReRe (GSPO) | **0.1266** | **0.1106** | **0.1458** | **0.1185** | 0.1716 | **0.1269** |

Table 9: Performance of ReRe with different training objectives.

We can observe that ReRe maintains high performance with different training algorithms, further indicating its generality. The design of RL loss tailored for recommendation scenarios tends to be a promising research direction.

# F  POPULARITY BIAS

The popularity bias issue is a crucial challenge in recommender systems. Following previous representative works discussing popularity bias (Liao et al., 2024a; Lichtenberg et al., 2024), we define the popularity bias as the difference between the average popularity level of recommended items and the average popularity level of the user's historical items.

$$\text{bias}_{\text{pop}}(u, \mathcal{I}_t) = \sum_{i_t \in \mathcal{I}_t} \frac{\text{LogPop}(i_t)}{|\mathcal{I}_t|} - \sum_{k=1}^{n} \frac{\text{LogPop}(i_k)}{n}, \quad (26)$$

where $i_t$ denotes the model-predicted item, $\{i_1 \cdots i_k\}$ denotes the interaction history of user $u$, and $|\mathcal{I}_t|$ represents the size of predicted item list $\mathcal{I}_t$.

The popularity bias of $D^3$ and ReRe on Amazon Industrial and Toys datasets is reported in Table 10. It can be observed that ReRe is less biased to popular items than baseline methods like $D^3$, which demonstrate additional advantages of ReRe in mitigating popularity bias.

Table 10: Popularity bias comparison between ReRe and $D^3$.

|  | Amazon Industrial | Amazon Toys |
|---|---|---|
| $D^3$ | 0.51 | 0.36 |
| ReRe | **-0.17** | **-0.28** |

# G  ITEM REPRESENTATION

Item representation is an important component of generative recommenders. Besides general textual titles, there are recommendation-tailored item representation paradigms. In this section, we explore the performance of ReRe across two additional representative item representation paradigms:

- Item collaborative embeddings (ID embedding): Following Liao et al. (2024b); Li et al. (2023), we concatenate textual item titles and item collaborative embeddings. Specifically, the item collaborative embeddings are sampled from SASRec models and mapped to the token embedding space of LLMs through a two-layer MLP.The parameters of SASRec are frozen, with the others being updated during the training process.

- Semantic IDs (SID): Following Zheng et al. (2023), we construct the SFT training dataset composed of title-SID alignment tasks and recommendation tasks. But the RL training dataset contains only recommendation data, and we adopt the SFT model as the initial model of ReRe for SID knowledge cold-start. The SIDs are obtained from RQ-VAE (Zeghidour et al., 2022).

The performance comparisons on Amazon Industrial dataset between $D^3$ and ReRe across ID embeddings and Semantic IDs are presented in Table 11 and Table 12, respectively. ReRe maintains effectiveness across different item representations.

Table 11: Performance comparison with ID embeddings.

|  | H@3 | N@3 | H@5 | N@5 | H@10 | N@10 |
|---|---|---|---|---|---|---|
| $D^3$ + ID emb | 0.1074 | 0.0944 | 0.1242 | 0.1013 | 0.1542 | 0.1110 |
| ReRe (base) + ID emb | **0.1231** | **0.1064** | **0.1430** | **0.1146** | **0.1763** | **0.1253** |

Table 12: Performance comparison with Semantic IDs.

|  | H@3 | N@3 | H@5 | N@5 | H@10 | N@10 |
|---|---|---|---|---|---|---|
| $D^3$ + SID | 0.0924 | 0.0808 | 0.1057 | 0.0863 | 0.1374 | 0.0965 |
| ReRe (SFT) + SID | **0.1048** | **0.0928** | **0.1229** | **0.1002** | **0.1562** | **0.1109** |

Table 13: Training and inference costs of ReRe across textual item titles and SIDs.

|  | Item Title | SID |
|---|---|---|
| Training | 2.5 h | 0.5 h |
| Inference | 7 min | 2 min |

Furthermore, representing items with SIDs substantially decreases both the training and inference costs depicted in Table 13 [8]. The high training and inference efficiency shows further potential and viability of SID-based ReRe.

# H  ANALYSIS ON $\beta$

We train ReRe with different values of $\beta$ in $\{1e-1, 1e-2, 1e-3, 1e-4\}$. The results are provided in Figure 8c, which indicates that generally a smaller value of $\beta$ tends to better adapt LLM-based recommenders. When $\beta$ is set too high, the KL divergence may impose too many restrictions on the model update. On the other hand, when the value of $\beta$ is too low, the learning process may become over-aggressive and cause a counterproductive result.

---

[8]The model is trained using 8 NVIDIA H20 GPUs on the Amazon Industrial dataset.

## I  USE OF LLMs

We primarily utilized LLMs for paper polishing and programming assistance. Specifically, for paper writing, we employed LLMs (*e.g.,* GPT-5) to refine sentence fluency as well as provide suggestions on precise and appropriate expression. All the modifications generated by LLMs were carefully reviewed by the authors to ensure that the final content faithfully reflects the authors' intended ideas. For code generation, LLMs (*e.g.,* Github Copilot) were mainly used to generate some code snippets, which were subsequently adapted and integrated into the implementation framework by the authors. In both paper polishing and programming assistance, the authors were in full control, with LLMs serving solely as tools to improve efficiency.

