# OpenReview forum: "Reinforced Preference Optimization for Recommendation"
_ICLR.cc/2026/Conference — Submitted to ICLR 2026_

### Official Review · Reviewer_yVsc · 2025-10-16

**Soundness:** 3
**Presentation:** 3
**Contribution:** 3
**Rating:** 6
**Confidence:** 3

**Summary:**

The paper proposes Reinforced Preference Optimization (RPO), a reinforcement learning–based extension of Direct Preference Optimization (DPO) for recommendation tasks. Unlike standard DPO, which aligns models to short-term revealed preferences, RPO introduces a utility critic that estimates long-term user satisfaction, combining it with reward modeling and policy updates. The framework alternates between preference optimization (based on pairwise comparisons) and utility reinforcement (based on delayed feedback).

**Strengths:**

The overall idea of integrating a reinforcement objective into DPO is reasonable and grounded in established RL principles. However, the derivation of the “utility critic” and the way it interacts with the preference policy is somewhat heuristic. The connection to standard RL formulations (e.g., Q-learning or advantage functions) is not rigorously developed.

**Weaknesses:**

The paper claims that reinforcement learning with verifiable rewards (RLVR) “naturally” addresses implicit reward issues in LLM-based recommenders, but this claim is not rigorously justified.

I am wondering what is verifiable reward? what will be the difference between verifiable reward and traditional reward in DL/RL?

The core of ReRe—beam search for sampling and ranking-based auxiliary rewards—is an extension of existing DPO and GRPO concepts. There is no much novel optimization algorithm or unique reward modeling principle. My main concern is that the proposed method primarily combines existing ideas (beam search, ranking loss, constrained decoding) with limited algorithmic innovation.

**Questions:**

Please refer to weakness part.

---

> ### Author Response · Authors · 2025-11-22
>
> We sincerely thank the reviewer for the feedback regarding how RLVR addresses implicit reward issues, the difference between verifiable rewards and traditional rewards, and the algorithmic innovation of ReRe. We address your comments in a point-by-point manner.
>
>
>
> > **How RLVR addresses implicit reward issues.** — "The paper claims that reinforcement learning with verifiable rewards (RLVR) “naturally” addresses implicit reward issues in LLM-based recommenders, but this claim is not rigorously justified."
>
> Thank you for giving us this opportunity to justify the merits of RLVR compared with previous implicit-reward-based methods for LLM-based recommender fine-tuning, which benefits the clarity of our paper. To make this point clearer, we elaborate on how RLVR addresses implicit reward issues conceptually and empirically.
>
> The implicit reward is broadly leveraged in direct preference alignment methods like DPO [1], which takes the following form:
> $$
> \hat{r}(e_t,x_u)=\beta\log\frac{\pi_\theta(e_t|x_u)}{\pi_{\rm ref}(e_t|x_u)},
> $$
> where $e_t$ denotes the tiltle of item $i_t$ and $x_u$ denotes the prompt bearing the interaction history of user $u$. DPO and its variants enlarge the margin between the preferred item title $e_p$ and the dis-preferred item title $e_d$, defined as $\hat{r}(e_p,x_u)-\hat{r}(e_d,x_u)$.
>
> However, the implicit reward is inconsistent with the recommendation accuracy. Namely, a larger implicit reward margin or a lower DPO loss does not indicate a better recommendation performance, similar to the reward over-optimization observed in the field of general-purpose LLMs [2,3].
>
> Empirically, we train S-DPO [4] on the Amazon Industrial dataset. Let $\mathcal{E}_d$ represent the titles of negative items. The average implicit reward margins $\text{Ave}=\frac{1}{|\mathcal{E}d|}\sum{e_d\in \mathcal{E}_d} (\hat{r}(e_p,x_u)-\hat{r}(e_d,x_u))$ across the negative items and recommendation performance (HR@10, NDCG@10) on the validation set of each checkpoint is presented below:
>
> **Dynamics of Recommendation Perfomance and Average Implicit Reward Margin**
>
> |                     | HR@10      | NDCG@10    | Ave Reward Margin |
> | ------------------- | ---------- | ---------- | ----------------- |
> | checkpoint step 57  | **0.1456** | **0.1111** | 0.1439            |
> | checkpoint step 114 | 0.1383     | 0.1063     | 0.4751            |
> | checkpoint step 171 | 0.1419     | 0.1079     | 0.5543            |
> | checkpoint step 228 | 0.1417     | 0.1078     | **0.5667**        |
>
> The results validate that the implicit reward margin is not well aligned with the recommendation accuracy.
>
> In contrast, the rule-based reward adopted in RLVR helps mitigate this misalignment problem. Specifically, the rule-based reward stems directly from the user feedback, defined as follows:
>
> - $r(e,x_u)= 0,e \ne e_p $,
> - $r(e,x_u) =  1,e=e_p $,
>
> which assigns the reward 1 if and only if the recommended item is the preferred item.
>
> We also compare the average rule-based reward with recommendation accuracy on the validation set of each checkpoint from ReRe.
>
> **Dynamics of Recommendation Perfomance and Average Rule-based Reward**
>
> |                      | HR@10      | NDCG@10    | Ave Rule-based Reward |
> | -------------------- | ---------- | ---------- | --------------------- |
> | checkpoint step 452  | 0.1461     | 0.1034     | 0.0110                |
> | checkpoint step 905  | 0.1567     | 0.1164     | 0.0116                |
> | checkpoint step 1358 | **0.1637** | 0.1253     | **0.0118**            |
> | checkpoint step 1811 | 0.1602     | 0.1259     | 0.0117                |
> | checkpoint step 2264 | 0.1602     | **0.1270** | 0.0116                |
>
> From the table above, the rule-based reward is generally more consistent with the recommendation accuracy, which means that the rule-based reward serves as a better-aligned indicator of recommendation performance.
>
> Therefore, from both conceptual and empirical perspectives, the rule-based reward in RLVR mitigates the reward-performance misalignment problems and thereby addresses implicit reward issues in direct preference alignment methods for LLM-based recommenders.
>
> [1] Rafailov R, Sharma A, Mitchell E, et al. Direct preference optimization: Your language model is secretly a reward model[J]. Advances in neural information processing systems, 2023, 36: 53728-53741.
>
> [2] Rafailov R, Chittepu Y, Park R, et al. Scaling laws for reward model overoptimization in direct alignment algorithms[J]. Advances in Neural Information Processing Systems, 2024, 37: 126207-126242.
>
> [3] D'Oosterlinck K, Xu W, Develder C, et al. Anchored preference optimization and contrastive revisions: Addressing underspecification in alignment[J]. Transactions of the Association for Computational Linguistics, 2025, 13: 442-460.
>
> [4] Chen Y, Tan J, Zhang A, et al. On softmax direct preference optimization for recommendation[J]. Advances in Neural Information Processing Systems, 2024, 37: 27463-27489.

---

> ### Author Response · Authors · 2025-11-22
>
> > **The definition of verifiable rewards and the difference between verifiable rewards and traditional rewards.** — "I am wondering what is verifiable reward? what will be the difference between verifiable reward and traditional reward in DL/RL?"
>
> We appreciate your question, and we would like to clarify the definiation of verifiable rewards and  difference between verifiable rewards and traditional rewards.
>
> Verifiable rewards are rewards computed by rule-based functions [1,2], such as binary accuracy rewards. In contrast, traditional rewards are typically from external reward models, such as those trained on pairwise preference data in reinforcement learning from human feedback (RLHF) [3,4].
>
> In the context of generative recommendation, the verifiable reward is 1 if and only if the predicted item is exactly the target item preferred by the user. Otherwise, the reward is 0. Traditional rewards are computed based on external models, such as the collaborative reward given by a collaborative model like SASRec [5]. The performances of these rewards on Amazon Industrial dataset are as follows.
>
> **Performance Comparison Between Verifiable Reward and Collaborative Reward**
>
> |                      | H@5        | N@5        | H@10       | N@10       |
> | -------------------- | ---------- | ---------- | ---------- | ---------- |
> | verifiable reward    | **0.1447** | **0.1171** | **0.1707** | **0.1256** |
> | collaborative reward | 0.0540     | 0.0296     | 0.0889     | 0.0414     |
>
> This performance comparison demonstrates the superiority of the verifiable reward in recommendation.
>
> For further exploration, we analyze the alignment between each reward and the recommendation accuracy. As depicted in Figure 3 in our manuscript, **the dynamics of the verifiable reward show excellent alignment with the recommendation accuracy**, **while a higher traditional reward does not ensure a better performance**. Similar over-optimization problems have also been observed in RLHF [6], where the reward increases but the performance drops. On the other hand, introduction of the verifiable reward can mitigate the reward hacking phenomenon [7, 8].
>
>
>
> [1] Guo D, Yang D, Zhang H, et al. Deepseek-r1: Incentivizing reasoning capability in llms via reinforcement learning[J]. arXiv preprint arXiv:2501.12948, 2025.
>
> [2] Wang Y, Yang Q, Zeng Z, et al. Reinforcement learning for reasoning in large language models with one training example[J]. arXiv preprint arXiv:2504.20571, 2025.
>
> [3] Ouyang L, Wu J, Jiang X, et al. Training language models to follow instructions with human feedback[J]. Advances in neural information processing systems, 2022, 35: 27730-27744.
>
> [4] Ziegler D M, Stiennon N, Wu J, et al. Fine-tuning language models from human preferences[J]. arXiv preprint arXiv:1909.08593, 2019.
>
> [5] Kang W C, McAuley J. Self-attentive sequential recommendation[C]//2018 IEEE international conference on data mining (ICDM). IEEE, 2018: 197-206.
>
> [6] Gao L, Schulman J, Hilton J. Scaling laws for reward model overoptimization[C]//International Conference on Machine Learning. PMLR, 2023: 10835-10866.
>
> [7] Yu Q, Zhang Z, Zhu R, et al. Dapo: An open-source llm reinforcement learning system at scale[J]. arXiv preprint arXiv:2503.14476, 2025.
>
> [8] Xie T, Gao Z, Ren Q, et al. Logic-rl: Unleashing llm reasoning with rule-based reinforcement learning[J]. arXiv preprint arXiv:2502.14768, 2025.

---

> ### Author Response · Authors · 2025-11-22
>
> > **The algorithmic innovation of ReRe.** — "The core of ReRe—beam search for sampling and ranking-based auxiliary rewards—is an extension of existing DPO and GRPO concepts. There is no much novel optimization algorithm or unique reward modeling principle. My main concern is that the proposed method primarily combines existing ideas (beam search, ranking loss, constrained decoding) with limited algorithmic innovation."
>
> Thank you for your valuable comment. To address your concern, we summarize the main algorithmic innovation of ReRe.
>
> RLVR [1] is broadly adopted and proved effective in training large reasoning models [2]. Furthermore, the rule-based reward possesses the advantageous property of mitigating reward hacking [3]. However, due to the unique characteristics of generative recommendation, **directly utilizing plain RLVR can not yield desirable performance**. The performance of plain GRPO on the Amazon Industrial dataset is presented below.
>
> **Performance Comparison among plain RLVR and Other Methods**
>
> |              | H@3        | N@3        | H@5        | N@5        | H@10       | N@10       |
> | ------------ | ---------- | ---------- | ---------- | ---------- | ---------- | ---------- |
> | $\text{D}^3$ | 0.1024     | 0.0911     | 0.1213     | 0.0989     | 0.1500     | 0.1082     |
> | plain RLVR   | 0.1046     | 0.0908     | 0.1238     | 0.0987     | 0.1452     | 0.1056     |
> | ReRe (base)  | **0.1222** | **0.1079** | **0.1447** | **0.1171** | **0.1707** | **0.1256** |
>
> It can be observed that **the performance of plain RLVR is only on par with baseline methods like $\text{D}^3$**. There lacks an effective paradigm to adapt RLVR to generative recommendation.
>
> To address this challenge, different from the common repeatedly sampling strategy, **ReRe is the first to introduce constrained beam search into RLVR to ensure the diversity of negative items**, which is hardly explored by previous works. In addition, **ReRe innovatively proposes the ranking reward modeling** that penalizes hard negatives more to inject finer-grained ranking information. **The ranking reward constitutes a novel reward mechanism tailored for recommendation.** The empirical results further validates the effectiveness of ReRe’s paradigm design.
>
> Empirically, **we also explore the possible extentension on ReRe**, including different reward designs, sampling strategies, numbers of generations, base models, item representations and optimization objectives, which contributes to the future design of RLVR paradigms for LLM-based recommenders.
>
>
>
> [1] Guo D, Yang D, Zhang H, et al. Deepseek-r1: Incentivizing reasoning capability in llms via reinforcement learning[J]. arXiv preprint arXiv:2501.12948, 2025.
>
> [2] Yu Q, Zhang Z, Zhu R, et al. Dapo: An open-source llm reinforcement learning system at scale[J]. arXiv preprint arXiv:2503.14476, 2025.
>
> [3] Xie T, Gao Z, Ren Q, et al. Logic-rl: Unleashing llm reasoning with rule-based reinforcement learning[J]. arXiv preprint arXiv:2502.14768, 2025.

---

### Official Review · Reviewer_Sk62 · 2025-10-27

**Soundness:** 3
**Presentation:** 3
**Contribution:** 3
**Rating:** 6
**Confidence:** 4

**Summary:**

This paper proposes ReRe to fix two flaws of LLM-based generative recommenders: poor high-quality negative modeling and reliance on implicit rewards. ReRe uses constrained beam search to improve sampling efficiency/diversify hard negatives and combines rule-based with ranking rewards for finer supervision.

**Strengths:**

1. ReRe effectively addresses the two key flaws of LLM-based generative recommenders (insufficient high-quality negative modeling and reliance on implicit rewards) by integrating constrained beam search (for improving sampling efficiency and diversifying hard negatives) and a combined reward (rule-based accuracy + auxiliary ranking rewards), directly tackling the unique generation space and sparse supervision challenges of RLVR adaptation .

2. The study uses three real-world datasets (Amazon Toys, Amazon Industrial, Yelp) and compares ReRe with diverse baselines.

3. ReRe maintains robust performance across different backbone models (Qwen2.5-1.5B, Gemma-2B, Qwen2.5-7B) and initialization methods (Base, SFT).

**Weaknesses:**

1. There is a lack of training-time efficiency comparisons with other generative recommendation methods (e.g., those that do not use reinforcement learning) as well as with traditional methods.

2. The dataset information in Table 5 is not clearly described, and the experiments rely exclusively on relatively small-scale datasets.

**Questions:**

1. In Table 5, do the numbers for “Tran” refer to the number of interactions or the number of users?

2. Can this method be combined with generative recommendation approaches (e.g., TIGER)? A semantic ID–based generative paradigm appears more practically viable, whereas relying on text prompts may constrain inference efficiency.

---

> ### Author Response · Authors · 2025-11-22
>
> We are grateful for your insightful comments on the efficiency comparison, dataset information and scale, as well as the integration of SIDs.
> We will address your comments point by point.
>
>
>
> > **Weakness 1: Lack of efficiency comparison.** — "There is a lack of training-time efficiency comparisons with other generative recommendation methods (e.g., those that do not use reinforcement learning) as well as with traditional methods."
>
> We highly appreciate the reviewer for pointing out the lack of a training efficiency comparison.
> To address your concerns, we provide **a detailed analysis of the training costs (GPU$\times$hours)** of our method, ReRe, against various baselines.
> We have **incorporated this comparison and discussion** into the Implementation Details subsection of the **Appendix** in the revised manuscript.
>
> The table below presents the training costs for ReRe, alongside generative methods—specifically, Preference Alignment (S-DPO [1]) and SFT-based (BigRec [2])—and a traditional baseline (SASRec [3]).
>
> **Note on Fairness**: For an equitable comparison, the reported costs for ReRe and S-DPO include their respective initial Supervised Fine-Tuning (SFT) training stage, as they are initialized with SFT models. Key hyperparameters were also aligned (e.g., ReRe's group size was set to 16, and S-DPO's number of negative items was set to 15).
>
> **Training Costs (GPU$\times$hours) of Different Methods across Different Datasets.**
>
> |            | ReRe | S-DPO | BigRec | SASRec |
> | ---------- | ---- | ----- | ------ | ------ |
> | Industrial | 20   | 23    | 1.2    | 0.042  |
> | Toys       | 40   | 48    | 4.0    | 0.22   |
> | Yelp       | 30   | 38    | 2.3    | 0.059  |
>
> It clearly shows that Preference Alignment methods (ReRe and S-DPO), by nature, incur significant overhead compared to simpler SFT or traditional methods due to mechanisms like on-policy or off-policy gradient computation.
> Furthermore, any generative model based on LLMs, such as BigRec, will inherently demand greater computational resources than traditional methods like SASRec.
>
> However, we wish to emphasize two critical points demonstrating the **efficiency and value** of ReRe:
>
> - **Efficiency advantage over peer preference alignment method.**
>   ReRe consistently demonstrates lower training costs than S-DPO.
>   This validates that ReRe effectively achieves preference alignment through beam search and constrained decoding instead of relying on costly off-policy preference gradient computation.
>
> - **Efficiency-performance trade-off of ReRe.**
>   Our paper demonstrates that ReRe's investment in computation yields substantial returns.
>   As shown in Table 1 of the main manuscript, ReRe achieves 10% - 30% relative improvement over S-DPO, and sometimes more than 110% relative improvement over SASRec.
>
> - **Potential for efficiency of ReRe.**
>   Thanks for your later question 2 (which we will address formally later), we confirm that utilizing SID can significantly reduce the computational costs of ReRe, further enhancing its practical deployability.
>
> [1] Chen Y, Tan J, Zhang A, et al. On softmax direct preference optimization for recommendation[J]. Advances in Neural Information Processing Systems, 2024, 37: 27463-27489.
>
> [2] Bao K, Zhang J, Wang W, et al. A bi-step grounding paradigm for large language models in recommendation systems[J]. ACM Transactions on Recommender Systems, 2025, 3(4): 1-27.
>
> [3] Kang W C, McAuley J. Self-attentive sequential recommendation[C]//2018 IEEE international conference on data mining (ICDM). IEEE, 2018: 197-206.

---

> ### Author Response · Authors · 2025-11-22
>
> > **Weakness 2: More detailed data information and experiments on a larger-scale dataset.** — "The dataset information in Table 5 is not clearly described, and the experiments rely exclusively on relatively small-scale datasets."
>
> Thank you for your feedback. A clearer description of data information is presented below, and **we have replaced Table 5 with this table in our revised manuscript.**
>
> **Data Information of Datasets.**
>
> |            | Users  | Items  | Density | Interactions | Train   | Valid  | Test   |
> | ---------- | ------ | ------ | ------- | ------------ | ------- | ------ | ------ |
> | Toys       | 22,158 | 11,251 | 0.065%  | 140,943      | 112,754 | 14,094 | 14,095 |
> | Industrial | 7,694  | 3,686  | 0.187%  | 45,324       | 36,259  | 4,532  | 4,533  |
> | Yelp       | 10,635 | 8,785  | 0.115%  | 96,372       | 77,097  | 9,637  | 9,638  |
>
> In terms of the relatively small data scales, we conduct additional experiments on a larger-scale dataset. Specifically, we sample interactions from October 2015 to November 2018 from the Amazon Toys dataset, with the information of this large dataset (referred to as "Toys-Large") presented below.
>
> **Data Information of Larger-scale Dataset.**
>
> |            | Users  | Items  | Density | Interactions | Train   | Valid  | Test   |
> | ---------- | ------ | ------ | ------- | ------------ | ------- | ------ | ------ |
> | Toys-Large | 72,947 | 32,301 | 0.025%  | 515,568      | 412,454 | 51,557 | 51,557 |
>
> We train $\text{D}^3$ and ReRe on this dataset, following the implementation details in our manuscript. The empirical results are presented below, indicating that ReRe remains effective on the larger-scale dataset.
>
> **Performance on Larger-scale Dataset.**
>
> |              | H@3        | N@3        | H@5        | N@5        | H@10       | N@10       |
> | ------------ | ---------- | ---------- | ---------- | ---------- | ---------- | ---------- |
> | $\text{D}^3$ | 0.0574     | 0.0485     | 0.0690     | 0.0533     | 0.0885     | 0.0596     |
> | ReRe (SFT)   | **0.0614** | **0.0524** | **0.0736** | **0.0575** | **0.0913** | **0.0632** |

---

> ### Author Response · Authors · 2025-11-22
>
> > **Question 1: Ambiguity of the numbers in Table 5.** — "In Table 5, do the numbers for “Tran” refer to the number of interactions or the number of users?"
>
> We apologize for the confusion of the numbers in Table 5. The numbers for “Train”, “Valid”, and “Test” in Table 5 refer to the number of interactions in the training, validation, and test datasets. Thanks for your question, and we have updated Table 5 in our manuscript for calrification.
>
>
>
> > **Question 2: Further exploration on SIDs.** — "Can this method be combined with generative recommendation approaches (e.g., TIGER)? A semantic ID–based generative paradigm appears more practically viable, whereas relying on text prompts may constrain inference efficiency."
>
> Thank you for your insightful suggention. The Semantic ID (SID) representation is an important item representation paradigm for recommendation, which encodes dense textual embeddings into discrete semantic IDs and achieves higher training and inference efficiency than textual title representations. We explore the performance and efficiency of ReRe when combined with SIDs, and **we have included the exploaration on SIDs in the Appendix**.
>
> There are many works adopting SIDs for item representations, such as TIGER [1] and LC-Rec [2]. We follow the paradigm of LC-Rec since it adapts SIDs to LLM-based recommenders. Specifically, we train Qwen2-0.5B on two categories of tasks through SFT : (1) SID-alignment tasks to inject title-SID mapping correlations into LLMs; (2) SID-recommendation tasks where LLMs are provided with SID historical interactions and required to directly generate target SIDs, both with SIDs from RQ-VAE [3]. On the other hand, RL training dataset contains only recommendation data, and ReRe is therefore initialized with the SFT model for further ranking capability enhancement. We also implement $\text{D}^3$ [4] under this setting as a baseline method. The empirical comparison is presented below.
>
> **Performance Comparison with Semantic IDs.**
>
> |                    | H@3        | N@3        | H@5        | N@5        | H@10       | N@10       |
> | ------------------ | ---------- | ---------- | ---------- | ---------- | ---------- | ---------- |
> | $\text{D}^3$﻿ + SID | 0.0924     | 0.0808     | 0.1057     | 0.0863     | 0.1374     | 0.0965     |
> | ReRe (SFT) + SID   | **0.1048** | **0.0928** | **0.1229** | **0.1002** | **0.1562** | **0.1109** |
>
> The results **demonstrate the effectiveness of ReRe when combined with SID representation**. Furthermore, **representing items with SIDs substantially decreases both the training and inference costs**. When trained on 8 NVIDIA H20 GPUs, the time costs of ReRe with item titles and SIDs are in the following table. **The high training and inference efficiency shows further potential and viability of SID-based ReRe.**
>
> **Training and Inference Costs of ReRe across Different Item Representations.**
>
> |           | Item Title | SID   |
> | --------- | ---------- | ----- |
> | Training  | 2.5 h      | 0.5 h |
> | Inference | 7 min      | 2 min |
>
> [1] Rajput S, Mehta N, Singh A, et al. Recommender systems with generative retrieval[J]. Advances in Neural Information Processing Systems, 2023, 36: 10299-10315.
>
> [2] Zheng B, Hou Y, Lu H, et al. Adapting large language models by integrating collaborative semantics for recommendation[C]//2024 IEEE 40th International Conference on Data Engineering (ICDE). IEEE, 2024: 1435-1448.
>
> [3] Zeghidour N, Luebs A, Omran A, et al. Soundstream: An end-to-end neural audio codec[J]. IEEE/ACM Transactions on Audio, Speech, and Language Processing, 2021, 30: 495-507.
>
> [4] Bao K, Zhang J, Zhang Y, et al. Decoding matters: Addressing amplification bias and homogeneity issue for llm-based recommendation[J]. arXiv preprint arXiv:2406.14900, 2024.

---

> > ### Comment · Reviewer_Sk62 · 2025-11-27
> >
> > Thanks for authors' detailed response, my main concerns have been addressed, so I will keep my score.

---

### Official Review · Reviewer_u1B8 · 2025-10-30

**Soundness:** 4
**Presentation:** 3
**Contribution:** 3
**Rating:** 6
**Confidence:** 4

**Summary:**

The authors have addressed reinforcement learning in generative recommenders. The authors found that existing works often rely on implicit rewards and ignore high-quality negative modeling. To address these two problems, this paper proposes to integrate the RLVR into the post-training of LLM-based recommenders. For adaptation, this paper proposed a constrained beam search and an augmenting rule-based accuracy reward. The extensive experiments have validated the effectiveness of the proposed method.

**Strengths:**

+ S1. This paper is well-organized and -written, making it easy to follow.
+ S2. This paper is well-motivated.
+ S2. Extensive experiments have been conducted.
+ S3. The code is released, making it easy to reproduce.

**Weaknesses:**

- W1. Some up-to-date papers were ignored by this paper. For example, LatentR3[1] also adopted the GRPO algorithm. What's the difference between ReRe and previous works?
- W2. It is better to further investigate the generality of the proposed ReRe. This paper has investigated the LLM-based recommender with the input of item titles, but how about the one with item ID, such as E4SRec or LLaRA?



[1]. Zhang, Yang, et al. "Reinforced Latent Reasoning for LLM-based Recommendation." *arXiv preprint [arXiv:2505.19092](https://arxiv.org/abs/2505.19092)* (2025).



[2]. Li, Xinhang, et al. "E4srec: An elegant effective efficient extensible solution of large language models for sequential recommendation." *arXiv preprint [arXiv:2312.02443](https://arxiv.org/abs/2312.02443)* (2023).



[3]. Liao, Jiayi, et al. "Llara: Large language-recommendation assistant." *Proceedings of the 47th International ACM SIGIR Conference on Research and Development in Information Retrieval*. 2024.

**Questions:**

All my questions have been included in the weakness section.

---

> ### Author Response · Authors · 2025-11-22
>
> Thank you for your valuable comments and suggestions on discussions of up-to-date papers and the generality of ReRe. We will provide detailed answers to your questions one by one.
>
>
>
> > **Weakness 1: Lack of discussions about up-to-date papers on RLVR for LLM-based recommenders.** — "Some up-to-date papers were ignored by this paper. For example, LatentR3[1] also adopted the GRPO algorithm. What's the difference between ReRe and previous works?"
>
> We appreciate your valuable comments on discussing up-to-date papers. To provide further clarification, we discuss the difference between our methods and other up-to-date papers and also conduct additional experiments for comparison.
>
> Some up-to-date papers also adopt RLVR in LLM-based recommendation fine-tuning. Besides Latent$\text{R}^3$[1] as you have mentioned, we have also surveyed other papers on RLVR for LLM-based recommenders, such as $\text{R}^2$ec [2] for a more comprehensive comparison.
>
> We want to emphasize that **the training paradigm of ReRe is different from these two methods** in terms of the sampling strategy, reward design, and the inference pipeline. Instead of introducing reasoning paths, ReRe adopts constrained beam search to sample diverse hard negative items directly. Furthermore, ReRe focuses on improving the model's ranking capability and incorporates the ranking reward for denser ranking information. **We have added the relevant citations and discussions about up-to-date papers on RLVR for LLM-based recommenders in the Related Work section.**
>
> The main differences among these works and ReRe are listed in the following table.
>
> **Comparison Between ReRe and Up-to-date papers on RLVR for recommendation.**
>
> | Method             | Reasoning         | Sampling                 | Reward                                | Inference                                           |
> | ------------------ | ----------------- | ------------------------ | ------------------------------------- | --------------------------------------------------- |
> | Latent$\text{R}^3$ | Latent Reasoning  | Reparameterization Trick | PPL on the Target Item                | Latent Reasoning then Generate                      |
> | $\text{R}^2$﻿ec     | Textual Reasoning | Common Sampling          | NDCG@K & Continuous Similarity Reward | Textual Reason then Retrieve with Last Hidden State |
> | ReRe               | -                 | Constrained Beam Search  | Rule-based Reward & Ranking Reward    | Directly Generate Items                             |
>
> In addition, **we implement Latent$\text{R}^3$ on the Amazon Indsutrial dataset.** Beneficial from the customized design for ranking information injection, ReRe successfully achieves superior performance.
>
> **Performance Comparison between Latent$\text{R}^3$ and ReRe.**
>
> |                    | H@3        | N@3        | H@5        | N@5        | H@10       | N@10       |
> | ------------------ | ---------- | ---------- | ---------- | ---------- | ---------- | ---------- |
> | $\text{D}^3$       | 0.1004     | 0.0901     | 0.1213     | 0.0983     | 0.1500     | 0.1079     |
> | Latent$\text{R}^3$ | 0.1118     | 0.0986     | 0.1297     | 0.1059     | 0.1553     | 0.1142     |
> | ReRe               | **0.1222** | **0.1079** | **0.1447** | **0.1171** | **0.1707** | **0.1256** |
>
> [1] Zhang Y, Xu W, Zhao X, et al. Reinforced Latent Reasoning for LLM-based Recommendation[J]. arXiv preprint arXiv:2505.19092, 2025.
>
> [2] You R, Li Y, Lin X, et al. R $^ 2$ ec: Towards Large Recommender Models with Reasoning[C]//The Thirty-ninth Annual Conference on Neural Information Processing Systems. 2025.

---

> ### Author Response · Authors · 2025-11-22
>
> > **Weakness 2: Further investigation into generality of ReRe.** — "It is better to further investigate the generality of the proposed ReRe. This paper has investigated the LLM-based recommender with the input of item titles, but how about the one with item ID, such as E4SRec or LLaRA?"
>
> Thank you for your insightful suggestion on further investigation into generality of ReRe across different item representations. Here we provide explore ReRe across two additional item representations.
>
> Item representation is an important component of generative recommenders. Besides general textual titles used in our manuscript, there are recommendation-tailored item representation paradigms, such as the collaborative embeddings (ID embeddings) and the semantic IDs (SIDs).
>
> We explore ReRe on these two item representation paradigms, specifically:
>
> 1. Item titles concatenated with item collaborative embeddings (following Llara [1] and E4SRec [2]).
> 2. Semantic IDs (following LC-Rec [3]).
>
> For collaborative embedding representation concatenation, we leverage item embeddings from SASRec [4] and map it to the token embedding space of LLMs through a two-layer MLP. The parameters of SASRec are frozen with the others being updated during the training process. The performance comparison on the Amazon Industrial dataset is presented below. **ReRe maintains the superiority over $\text{D}^3$ with item collaborative embeddings mapped to the language model space for collaborative knowledge instillation.**
>
> **Performance Comparison with Item ID Embeddings Integrated.**
>
> |                       | H@3        | N@3        | H@5        | N@5        | H@10       | N@10       |
> | --------------------- | ---------- | ---------- | ---------- | ---------- | ---------- | ---------- |
> | $\text{D}^3$ + ID emb | 0.1074     | 0.0944     | 0.1242     | 0.1013     | 0.1542     | 0.1110     |
> | ReRe (base) + ID emb  | **0.1231** | **0.1064** | **0.1430** | **0.1146** | **0.1763** | **0.1253** |
>
> For semantic ID representation, the items are represented as semantic IDs from RQ-VAE [5]. In order to help LLMs understand SIDs, the SFT training dataset comprises title-SID alignment tasks and recommendation tasks, but the RL training dataset contains only recommendation data. Therefore, we adopt the SFT model as the initial model of ReRe for SID knowledge cold-start. The performance on the Amazon Industrial dataset is as follows. **ReRe can also improve the model’s ability to understand and recommend SIDs.**
>
> **Performance Comparison with Semantic IDs.**
>
> |                    | H@3        | N@3        | H@5        | N@5        | H@10       | N@10       |
> | ------------------ | ---------- | ---------- | ---------- | ---------- | ---------- | ---------- |
> | $\text{D}^3$﻿ + SID | 0.0924     | 0.0808     | 0.1057     | 0.0863     | 0.1374     | 0.0965     |
> | ReRe (SFT) + SID   | **0.1048** | **0.0928** | **0.1229** | **0.1002** | **0.1562** | **0.1109** |
>
> These results further demonstrate the generality of ReRe across different item representation paradigms, and further explorations can be conducted on combining RLVR with various item representations. **We have also included the results across different item representations in the Appendix.**
>
> [1] Liao J, Li S, Yang Z, et al. Llara: Large language-recommendation assistant[C]//Proceedings of the 47th International ACM SIGIR Conference on Research and Development in Information Retrieval. 2024: 1785-1795.
>
> [2] Li X, Chen C, Zhao X, et al. E4srec: An elegant effective efficient extensible solution of large language models for sequential recommendation[J]. arXiv preprint arXiv:2312.02443, 2023.
>
> [3] Zheng B, Hou Y, Lu H, et al. Adapting large language models by integrating collaborative semantics for recommendation[C]//2024 IEEE 40th International Conference on Data Engineering (ICDE). IEEE, 2024: 1435-1448.
>
> [4] Kang W C, McAuley J. Self-attentive sequential recommendation[C]//2018 IEEE international conference on data mining (ICDM). IEEE, 2018: 197-206.
>
> [5] Zeghidour N, Luebs A, Omran A, et al. Soundstream: An end-to-end neural audio codec[J]. IEEE/ACM Transactions on Audio, Speech, and Language Processing, 2021, 30: 495-507.

---

> > ### Comment · Reviewer_u1B8 · 2025-11-27
> >
> > All my concerns have been well addressed, so I'd like to raise the score.

---

### Official Review · Reviewer_XSob · 2025-10-30

**Soundness:** 3
**Presentation:** 3
**Contribution:** 3
**Rating:** 6
**Confidence:** 3

**Summary:**

This paper addresses two major limitations of Reinforcement Learning with Verifiable Rewards (RLVR) in generative recommendation models. First, the unique generative space often produces invalid or duplicate items, reducing sampling efficiency. Second, since most items receive identical zero rewards, the ranking supervision signals become sparse. To overcome these issues, the authors propose Reinforced Preference Optimization for Recommendation (ReRe). Specifically, ReRe introduces a constrained beam search mechanism to improve sampling efficiency and increase the diversity of hard negative samples. In addition, it supplements rule-based accuracy rewards with auxiliary ranking rewards, enabling finer-grained supervision.

**Strengths:**

1.ReRe effectively addresses the challenge of hard negative sampling by introducing constrained beam search.

2.The use of ranking rewards alleviates the limitations of binary rule-based supervision.

3.The experimental results demonstrate solid performance and validate the method’s effectiveness.

4.The overall structure and logic of the paper are clear and well-organized.

**Weaknesses:**

1.For LLM-based recommender systems, prompt design is a crucial component, yet the paper does not discuss it.

2.Figure 2 fails to clearly illustrate ReRe’s contribution to negative sample sampling; further clarification or revision is needed.

3.Although constrained beam search mitigates the generation of invalid or duplicate items, ReRe may still be biased toward popular items due to the long-tail distribution, potentially overlooking less popular ones.

**Questions:**

See weakness

---

> ### Author Response · Authors · 2025-11-22
>
> We appreciate your comments on the prompt design, clarity of illustrations, and popularity bias. We will reply to each question individually.
>
>
>
> >  **Weakness 1: Lack of prompt design discussion. —** "For LLM-based recommender systems, prompt design is a crucial component, yet the paper does not discuss it."
>
> Thank you for your feedback on the prompt design discussion. For better clarification, we provide our prompt template and additional empirical results on prompt design.
>
> Our prompt design is based on the implementations of $\text{D}^3$ [1]. One of the prompt templates used in ReRe is provided below, and **we have added the full prompt design in the Methodology Details section of the Appendix.**
>
> > Below is an instruction that describes a task, paired with an input that provides further context. Write a response that appropriately completes the request.
> >
> >
> >
> > \### Instruction:
> >
> > Given a list of {*category*}s the user recently enjoyed, please write a new {*category*} that the user may buy.
> >
> >
> >
> > \### User Input:
> >
> > The user has browsed the following {*category*}s before: {*history*}.
> >
> >
> >
> > \### Response:
> >
> > {*output*}
>
> Here {*category*},  {*history*}, and {*output*} denote the item category, user interaction history, and the target item, respectively. Following Llara [2], we sample one instruction from 11 instructions during training to improve the training stability.
>
> Additionally, we explore the impact of prompt designs on the recommendation performance. The performance comparison is presented in the following table, where we compare three prompt design variations:
>
> 1. Original: the original prompt design.
> 2. One instruction: fixing the instruction during training and inference.
> 3. W/o instruction: eliminating texts preceding “### User Input”.
>
> **Performance of ReRe with Different Prompt Designs on Amazon Industrial.**
>
> | prompt design   | N@5    | H@5    | N@10   | H@10   |
> | --------------- | ------ | ------ | ------ | ------ |
> | original        | 0.1171 | 0.1447 | 0.1256 | 0.1707 |
> | one instruction | 0.1155 | 0.1418 | 0.1252 | 0.1719 |
> | w/o instruction | 0.1176 | 0.1441 | 0.1260 | 0.1701 |
>
> It can be observed that different prompt designs generally lead to different performances, but the influence is relatively slight. We have also surveyed papers regarding prompt design for LLM-based recommenders, which provide practical prompt selection guidelines [3] or convert user profiles and historical interactions into prompts [4]. We are happy to further discuss this with you.
>
>
>
> > **Weakness 2: Vagueness of Figure 2.** — "Figure 2 fails to clearly illustrate ReRe’s contribution to negative sample sampling; further clarification or revision is needed."
>
> We appreciate your suggestion on the illustration of the negative sampling distribution in Figure 2. The key contribution of ReRe to the negative sampling primarily lies in integrating constrained beam search into RLVR for LLM-based recommenders, which ensures the heterogeneity of items as well as the consistency between the distribution of negative items and the training policy.
>
> To better clarify the negative sampling strategy of ReRe, **we have revised Figure 2 and its caption to highlight that** **the generated items are sampled directly from the training policy $\pi_\theta$** **via constrained beam search, and are mutually distinct**. Combined with Figure 1 in our manuscript, we hope that our figures can more clearly illustrate ReRe's contribution to negative sampling.
>
>
>
> [1] Bao K, Zhang J, Zhang Y, et al. Decoding matters: Addressing amplification bias and homogeneity issue for llm-based recommendation[J]. arXiv preprint arXiv:2406.14900, 2024.
>
> [2] Liao J, Li S, Yang Z, et al. Llara: Large language-recommendation assistant[C]//Proceedings of the 47th International ACM SIGIR Conference on Research and Development in Information Retrieval. 2024: 1785-1795.
>
> [3] Kusano G, Akimoto K, Takeoka K. Are longer prompts always better? prompt selection in large language models for recommendation systems[J]. arXiv preprint arXiv:2412.14454, 2024.
>
> [4] Gao Y, Sheng T, Xiang Y, et al. Chat-rec: Towards interactive and explainable llms-augmented recommender system[J]. arXiv preprint arXiv:2303.14524, 2023.

---

> ### Author Response · Authors · 2025-11-22
>
> > **Weakness 3: Lack of popularity bias analysis.** — "Although constrained beam search mitigates the generation of invalid or duplicate items, ReRe may still be biased toward popular items due to the long-tail distribution, potentially overlooking less popular ones."
>
> Thank you for your insightful comment. To address your concern, we further analyze the popularity bias of ReRe.
>
> The popularity bias issue is a crucial challenge in recommender systems. There are some representative works [1,2] discussing the popularity bias of LLM-based recommenders. Following these works, we define the popularity bias as the difference between the popularity level of recommended items and the average popularity level of the user's historical items:
> $$
> bias_{pop}(u,i_t)=LogPop(i_t)-\sum_{k=1}^n\frac{LogPop(i_k)}{n},
> $$
>
> where $i_t$ denotes the model-predicted item and $\{i_1\cdots i_k\}$ denotes the interaction history of user $u$.
>
> Different from selecting one item from a candidate set in RosePO [2], the model is required to generate a predicted item list $I_t$  in our experimental setting. Thus, we regard the average popularity bias of predicted items as the popularity bias of the whole list, formulated as follows:
> $$
> bias_{pop}(u,I_t)=\sum_{i_t\in I_t} \frac{bias_{pop}(u,i_t)}{|I_t|}=\sum_{i_t\in I_t}\frac{LogPop(i_t)}{|I_t|}-\sum_{k=1}^n\frac{LogPop(i_k)}{n}
> $$
> Here, $|I_t|$ represents the size of predicted item list $I_t$ .
>
> The average popularity bias of $\text{D}^3$ [3] and ReRe on Amazon Industrial and Toys datasets is reported below. It can be observed that ReRe is less biased to popular items than baseline methods like $\text{D}^3$, which demonstrate **additional advantages of ReRe in mitigating popularity bias**. **We have incorporated this popularity bias analysis into the updated Appendix.** Further exploration can be conduted on designing new RLVR methods for popularity debias and the mechanism of how RLVR methods mitigate popularity bias. We would like to further discuss this with you.
>
> **Popularity Bias Comparison between $\text{D}^3$ and ReRe on Amazon Industrial and Toys datasets.**
>
> |              | Amazon Industrial | Amazon Toys |
> | ------------ | ----------------- | ----------- |
> | $\text{D}^3$ | 0.51              | 0.36        |
> | ReRe         | **-0.17**         | **-0.28**   |
>
>
>
> [1] Lichtenberg J M, Buchholz A, Schwöbel P. Large language models as recommender systems: A study of popularity bias[J]. arXiv preprint arXiv:2406.01285, 2024.
>
> [2] Liao J, He X, Xie R, et al. Rosepo: Aligning llm-based recommenders with human values[J]. arXiv preprint arXiv:2410.12519, 2024.
>
> [3] Bao K, Zhang J, Zhang Y, et al. Decoding matters: Addressing amplification bias and homogeneity issue for llm-based recommendation[J]. arXiv preprint arXiv:2406.14900, 2024.

---

> > ### Comment · Reviewer_XSob · 2025-11-26
> > **Response to Authors' Rebuttal**
> >
> > Thank you for the detailed responses, which have addressed most of my questions and concerns. I am willing to raise my score based on the clarifications provided.
> >
> > Before doing so, I would like to further understand one remaining issue regarding popularity bias: What proportion of popular items appears in the candidate list recommended by ReRe? Popular items are typically defined as those within the top 10% or top 20% in terms of interaction frequency in the dataset.

---

> ### Author Response · Authors · 2025-11-27
>
> Thank you for your continual feedback. We compute the proportions of top-10% popular items and top-20% popular items in the recommendation items from ReRe and $\text{D}^3$. The results on the test sets of Amazon Industrial and Toys datasets are reported below.
>
> **Proportion of Top-10% Popular Items in Recommendation Items.**
>
> |              | Amazon Industrial | Amazon Toys |
> | ------------ | ----------------- | ----------- |
> | ReRe         | **33.7%**         | **29.5%**   |
> | $\text{D}^3$ | 59.6%             | 52.6%       |
>
> **Proportion of Top-20% Popular Items in Recommendation Items**
>
> |      | Amazon Industrial | Amazon Toys |
> | ---- | ----------------- | ----------- |
> | ReRe | **48.3%**         | **43.2%**   |
> | $\text{D}^3$   | 74.8%             | 67.3%       |
>
> It can be observed that ReRe is less prone to recommending highly popular items, which further validates the ability of ReRe to mitigate popularity bias.

---

> > ### Comment · Reviewer_XSob · 2025-11-27
> > **Response to Authors' Rebuttal**
> >
> > Thank you for the detailed responses. I am willing to raise my score based on the clarifications provided.

---

### Author Response · Authors · 2025-12-04
**Summary of Rebuttal**

Below, we provide an overview of the **strengths** recognized by the reviewers, the **concerns**, and our **responses**.

---

**Strength**

1. **Effective method**: Reviewers $\color{blue}{\textbf{XSob}}$, $\color{red}{\textbf{u1B8}}$, $\color{green}{\textbf{Sk62}}$ and $\color{purple}{\textbf{yVsc}}$ recognize our method as **"effective"** and **"reasonable"**, especially the designs of the constrained beam search and the ranking reward (reviewers $\color{blue}{\textbf{XSob}}$ and $\color{green}{\textbf{Sk62}}$ ). Reviewer $\color{red}{\textbf{u1B8}}$ also notes that our method is "**well-motivated**".
2. **Solid performance**: Reviewers $\color{blue}{\textbf{XSob}}$ and $\color{green}{\textbf{Sk62}}$ think the performance of ReRe is **"solid"** and **"robust"**. Reviewer $\color{red}{\textbf{u1B8}}$ mentions that **"extensive experiments have been conducted"**.
3. **Clear writing**: Reviewers $\color{blue}{\textbf{XSob}}$ and $\color{red}{\textbf{u1B8}}$ both think our paper is "**well-organized**". The logic of our paper is **clear** (reviewer $\color{blue}{\textbf{XSob}}$ ) and **easy to follow** (reviewer $\color{red}{\textbf{u1B8}}$).

---

**Concerns and Responses**

1. **Popularity bias**: Reviewer $\color{blue}{\textbf{XSob}}$ raised a concern that ReRe might overly favor popular items.
   To solve their concerns, we conducted **additional empirical analyses** and found that **ReRe is less prone to recommending popular items** than the baselines.

2. **Different item representations**: Reviewers $\color{red}{\textbf{u1B8}}$ and $\color{green}{\textbf{Sk62}}$ suggested investigating ReRe under different item representations, including item collaborative embeddings and semantic IDs.
   Following their suggestions, we **added experiments** demonstrating the **effectiveness of ReRe with different item representations**, further strengthening the generality of the framework.
   In addition, we observed that when combined with semantic IDs, **the training and inference costs of ReRe can be significantly decreased, further enhancing its practical deployability**.

3. **Other details**: In response to reviewers' requests, we added more details and experimental results on **prompt design** (reviewer  $\color{blue}{\textbf{XSob}}$), **comparisons and discussions about up-to-date papers** (reviewer $\color{red}{\textbf{u1B8}}$), **dataset information and scale** (reviewer $\color{green}{\textbf{Sk62}}$) and **training cost comparison** (reviewer $\color{green}{\textbf{Sk62}}$). We also **modified our illustrations** (reviewer $\color{blue}{\textbf{XSob}}$) for better clarity, and **further specified the advantages of verifiable rewards over implicit rewards and traditional rewards** (reviewer $\color{purple}{\textbf{yVsc}}$).

We have **included these modifications** in our manuscript (rendered in blue in the PDF), and we believe our paper has been greatly improved with these meaningful comments.

We thank the reviewers once again for their valuable suggestions and ongoing engagement in the discussions.

Thanks and best regards,

Authors of paper 17548

---

### Author Response · Authors · 2025-12-04
**Summary of Rebuttal**

Dear Reviewers and Area Chair,

We would like to express our sincere appreciation to all the reviewers for their constructive and insightful reviews, as well as the continuous discussions, which have substantially improved our manuscript.

In this work, we propose **ReRe**, a simple yet effective adaptation of RLVR for recommendation.
ReRe introduces constrained beam search to incorporate diverse hard negatives during RLVR training, and a ranking reward to inject finer-grained ranking signals into LLM-based recommenders.

Our paper received initial scores of **6,6,6,6**.

During the discussion phase, we have "addressed most of the questions and concerns" of the reviewers $\color{blue}{\textbf{XSob}}$ and $\color{green}{\textbf{Sk62}}$. In addition, we have "**well addressed** all the concerns" of the reviewer $\color{red}{\textbf{u1B8}}$. Among the reviewers who have continuous feedback on our responses, Reviewer $\color{blue}{\textbf{XSob}}$ is **"willing to raise the score based on the clarifications provided"**, reviewer $\color{red}{\textbf{u1B8}}$ **"would like to raise the score"**, and Reviewer $\color{green}{\textbf{Sk62}}$ also keeps the original positive score. Although reviewer $\color{purple}{\textbf{yVsc}}$ has no further comments, he or she recognizes our idea as **"reasonable and grounded in established RL principles"**.

---

### Meta-Review · Area_Chair_XUH1 · 2025-12-27

**Summary:**

Here are the reviewers' concerns that informed my suggested decision for this paper.

1.Lack of popularity bias analysis, proposed by Reviewer XSob.

2.Lack of discussions with recent recommendation papers using GRPO, proposed by Reviewer u1B8.

3.The generalization ability of ReRe on Item ID or Semantic IDs, proposed by Reviewer u1B8.

4.The experiments rely exclusively on relatively small-scale datasets, proposed by Reviewer Sk62.

5.Limited algorithmic novelty, proposed by Reviewer yVsc.

This paper proposes applying the GRPO algorithm to LLM-based Recommender Systems. The proposed ReRe method introduces constrained beam search and rule-based accuracy rewards augmented with auxiliary ranking rewards, demonstrating performance improvements on several public datasets. Although reviewers provided generally positive scores, concerns remain regarding its high similarity to the existing method R²ec (NeurIPS 2025), a lack of comparative experiments with R²ec, and the limited algorithmic novelty compared with existing methods. Considering ICLR's high standards, I decided to reject this submission.

**Reviewer Concerns:**

1.Lack of popularity bias analysis, proposed by Reviewer XSob: During the rebuttal, the author defines the popularity bias, and compares ReRe with D^3 in terms of this metric, and show that that ReRe is less prone to recommending highly popular items. I believe this point is addressed by the rebuttal.

2.Lack of discussions with recent recommendation papers using GRPO, proposed by Reviewer u1B8: During the rebuttal, the author discuss the difference between ReRe and LatentR^3, R²ec (NeurIPS 2025). I find that the paper is quite similar with R²ec, both applying RLVR algorithms such as GRPO to recommendations. The main differences between R²ec and ReRe are just the sampling strategies and reward forms. Also, the author does not provide comparison results with R²ec. Thus, I think the point is still outstanding.

3.The generalization ability of ReRe on Item ID or Semantic IDs, proposed by Reviewer u1B8: During the rebuttal, the author provides new experiments that implement ReRe with Item ID and SIDs, which show that ReRe works for the new setting. I believe this point is addressed by the rebuttal.

4.The experiments rely exclusively on relatively small-scale datasets, proposed by Reviewer Sk62: During the rebuttal, the author provides new results on a larger version of Amazon Toys dataset. I believe this point is addressed by the rebuttal.

5.Limited algorithmic novelty, proposed by Reviewer yVsc: During the rebuttal, the author argues that ReRe proposes the constrained beam search and the ranking reward modeling. However, both are implementation tricks. R^2ec proposes to use NDCG as the reward function, but the paper does not discuss the difference of these rewards.  Thus, I think the point is still outstanding.

**Reviewer Scores:**

Reviewer XSob would increase his or her score from 6 if he or she has been able to participate fully in the discussion.

Reviewer u1B8 would increase his or her score from 6 if he or she has been able to participate fully in the discussion.

Reviewer Sk62  would keep his or her score to 6  if he or she has been able to participate fully in the discussion.

Reviewer yVsc would keep his or her score to 6  if he or she has been able to participate fully in the discussion.

---

### Decision · Program_Chairs · 2026-01-26

Reject